

# Systematic analysis of relative phase extraction in one-dimensional Bose gases interferometry

**Taufiq Murtadho[1⋆], Marek Gluza[1], Khatee Zathul Arifa[1,2],**
**Sebastian Erne[3], Jörg Schmiedmayer[3] and Nelly H.Y. Ng[1†]**

**1** School of Physical and Mathematical Sciences, Nanyang Technological University,
637371 Singapore, Republic of Singapore
**2** Department of Physics, University of Wisconsin-Madison, 53706 Madison, USA
**3** Vienna Center for Quantum Science and Technology, Atominstitut, TU Wien,
Stadionallee 2, 1020 Vienna, Austria

⋆ fiqmurtadho@gmail.com , † nelly.ng@ntu.edu.sg

## Abstract

Interference upon free expansion gives access to the relative phase between two interfering matter waves. Such measurements can be used to reconstruct the spatially-resolved relative phase, which is a key observable in many quantum simulations of quantum field theory and non-equilibrium experiments. However, in 1D systems, longitudinal dynamics is typically ignored in the analysis of experimental data. In our work, we give a detailed account of various effects and corrections that occur in finite temperatures due to longitudinal expansion. We provide an analytical formula showing a correction to the readout of the relative phase due to longitudinal expansion and mixing with the common phase. Furthermore, we numerically assess the error propagation to the estimation of the gases' physical quantities such as correlation functions and temperature. We also incorporate systematic errors arising from experimental imaging devices. Our work characterizes the reliability and robustness of interferometric measurements, directing us to the improvement of existing phase extraction methods necessary to observe new physical phenomena in cold-atomic quantum simulators.

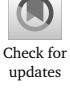

# 1 Introduction

Matter-wave interference highlights the quantum nature of matter, while simultaneously enabling ultra-precise sensors for metrology and providing a sensitive probe into the intricate many-body physics of ultracold quantum gases and quantum simulators [1–4]. A key technique for the latter is time-of-flight (TOF) measurements, where the quantum gas expands upon being released from the trap. If two such expanded clouds overlap, they form a matter-wave interference pattern that reveals the relative phase between the trapped clouds. If the expansion preserves local information, properties connected to the local relative phase in the original samples can be extracted.

This rationale has been extensively applied, particularly in 1D cold-atomic quantum field simulators, to investigate quench dynamics [5–8], pre-thermalization [9–12], area-law scaling of mutual information [13], and quantum thermodynamics [14, 15]. These applications leverage the statistical properties of local relative phases to deduce key physical quantities of the gas, such as temperature [16, 17], relaxation time scales [8, 18], the nature of excitations via full distribution functions [12, 19], and the quantum field theory description through correlation functions [20, 21]. In all of these investigations, measuring interference patterns after time-of-flight and then inferring the local relative phase fluctuation is an essential tool.

In this work, we perform a focused study on the TOF measurement of two parallel 1D Bose gases, going beyond the initial idealized reasoning in Refs. [22–24]. We systematically address various physical processes that can modify the interference patterns and thereby the extraction of the local relative phase. We assess the accuracy of the *decoding*, i.e. the inference of the relative phase in the trapped clouds from the observed interference patterns. A detailed and systematic analysis of the various effects influencing TOF measurement becomes indispensable when pushing further the analysis of low-dimensional many-body quantum systems and the quantum field simulators they enable.

To reliably extract the relative phase, we need an accurate understanding of the measurement dynamics. If the trap is switched off rapidly, the initial tight confinement in transverse directions leads to rapidly expanding density, which allows one to neglect the effect of inter-

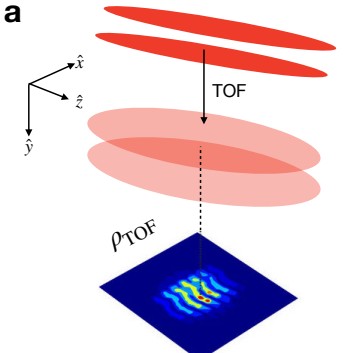

| Modelling assumptions | $\rho_{\text{TOF}}^{\perp}$ | $\rho_{\text{TOF}}$ |
|---|:---:|:---:|
| $G(z - z', t) \to \delta(z - z')$ | ✓ | ∼ |
| $a_s(t > 0) \to 0$ | ✓ | ✓ |
| $\hat{\psi}_{1,2} \sim \psi_{1,2}$ | ✓ | ✓ |
| $\sigma_0(z) \approx \sigma_0$ | ✓ | ✓ |
| $\omega_\perp t \gg 1$ | ✓ | ✓ |
| $d \ll \sigma_t$ | ✓ | ✓ |
| $\delta n_{1,2} \to 0$ | ∼ | ✓ |
| $n_1(z) = n_2(z) = n_0(z)$ | ∼ | ✓ |

Figure 1: (**a**) The setup schematics for relative phase measurement of parallel quasi-one dimensional Bose gases after time-of-flight (TOF) adapted from Ref. [11] (**b**) Comparison table for the assumptions used to derive different models for TOF density [Eqs. (4)-(5)]. The ∼ symbol means that the assumption can be relaxed in general.

actions during the expansion. Consequently, the dynamics is well approximated by a quench into free evolution [16, 24]. For 1D systems, such free expansion can be divided into expansion in the transversal directions (perpendicular to the length of the gas) and longitudinal direction (along the length of the gas). Although previous studies [22, 23] often neglect longitudinal expansion, recent theoretical works have started to address its significance [24]. In particular, they unveil new phenomena affecting the formation of interference patterns such as density ripples [16] and mixing with common (symmetric) phases [24]. A natural question then arises: how do these factors influence the relative phase extraction and the determination of the gases' physical properties? To the best of our knowledge, no systematic answer has been offered in the literature. This paper therefore aims to comprehensively address this question.

The paper is structured as follows: after this brief introduction, we summarize the developments in modelling TOF measurement dynamics for parallel 1D systems in Sec. 2. Then, in Sec. 3, we develop a perturbative theory for incorporating longitudinal dynamics and derive analytical expressions for the systematic readout errors in the extracted relative phase. Secs. 4-5 provide numerical analyses to assess the influence of the readout errors on the estimation of the various physical quantities of the gases, accounting for modelling errors (Sec. 4) and the effects of the experimental imaging system (Sec. 5). We conclude with a brief discussion and outlook in Sec. 6.

## 2 Free expansion dynamics of parallel 1D Bose gases

We consider a pair of parallel one-dimensional Bosonic gases of length $L$ extending along the $z$-axis (longitudinal axis) and separated by a distance $d$ along one of the transversal axes, e.g. the $x$-axis [Fig. 1a]. Let $\hat{\psi}_j(z)$ be the Bosonic field operator with subscripts $j = 1, 2$ indexing each gas. This operator can be decomposed as $\hat{\psi}_j(z) = e^{i\hat{\phi}_j(z)}\sqrt{\hat{n}_j(z)}$ with $\hat{n}_j(z)$ and $\hat{\phi}_j(z)$ being the density and phase operators. In this paper, we will use the semi-classical approximation [25] by replacing $\hat{\psi}_j(z)$ with a scalar field $\psi_j(z) = e^{i\phi_j(z)}\sqrt{n_j(z) + \delta n_j(z)}$ where $n_j(z)$ is the mean density, and $\delta n_j(z), \phi_j(z)$ are density and phase fluctuations respectively. The objective of 1D Bose gases interferometry is to measure the relative phase fluctuation $\phi_-(z) = \phi_2(z) - \phi_1(z)$. This can be achieved through TOF scheme, whereby the atomic cloud is imaged after being released and expanded for some time $t$. The image encodes information about the in-situ phase fluctuations in the resulting density interference patterns measured in experiments.

In the following, we assume the system to be initially in the quasi-1D regime, i.e. only occupying the Gaussian transverse ground state wavefunction [22, 26, 27]

$$\Psi_j(x,y,z,0) = \frac{1}{\sqrt{\pi\sigma_0^2(z)}} \exp\left(-\frac{(x \pm d/2)^2 + y^2}{2\sigma_0(z)^2}\right)\psi_j(z), \tag{1}$$

where the right and left wells are assumed to be symmetric with respect to the origin. The Gaussian width $\sigma_0^2(z) = \sigma_0^2\sqrt{1 + 2a_s n_j(z)}$ depends on the scattering length $a_s$, the mean density $n_j(z)$, and the single-particle ground state width $\sigma_0 = \sqrt{\hbar/(m\omega_\perp)}$ given by the atomic mass $m$ and the transverse harmonic confinement frequency $\omega_\perp$. For the moment, we will ignore the radial broadening due to atomic repulsion such that the width $\sigma_0(z) \equiv \sigma_0$ is uniform along the condensate. We briefly discuss the effect of scattering in Sec. 6 and Appendix E.

We model TOF expansion as a ballistic expansion, without any external potential nor any interaction (i.e. $a_s(t > 0) = 0$). The latter is justified due to the fast decrease of interaction energy as a result of the rapid expansion of the gas in the tightly confined transverse directions characterized by $\omega_\perp$. Thus, for $t > 0$ the system is effectively governed by free particle dynamics [16, 24]

$$\Psi_j(\vec{r}, z, t) = \int d^2\vec{r}' \, dz' \, G(\vec{r} - \vec{r}', t)G(z - z', t)\, \Psi_j(\vec{r}', z', 0), \tag{2}$$

where $\vec{r}$ is a short-hand notation for the position vector in the transverse plane and $G(\xi, t) = \sqrt{m/(2\pi i \hbar t)}e^{-m\xi^2/2i\hbar t}$ is the free, single-particle Green's function. We also note that recent work [28, 29] have developed fast and efficient methods to numerically evaluate Eq. (2). In our analytical contributions, we make use of additional approximations to obtain a simplified analytical form of the time evolution. Thus, our results are complementary to that of Refs. [28, 29], while paving the way for a further systematic understanding of the TOF scheme.

As the gases expand, they start to overlap and coherently interfere. We are interested in the density image of the atomic cloud after interference as seen from the vertical direction ($y$-axis), i.e.

$$\rho_{\text{TOF}}(x, z, t) = \int dy \, |\Psi_1(\vec{r}, z, t) + \Psi_2(\vec{r}, z, t)|^2. \tag{3}$$

After substituting the time-evolved fields from Eq. (2) and applying the assumptions listed in Fig. 1b, one arrives at a simplified formula for the expanded density [22, 23]

$$\rho_{\text{TOF}}^\perp(x, z, t) = A(z, t)e^{-x^2/\sigma_t^2}\left[1 + C(z)\cos\left(kx + \phi_-(z)\right)\right], \tag{4}$$

where $\sigma_t = \sigma_0\sqrt{1 + \omega_\perp^2 t^2}$ is the expanded Gaussian width, $k(t) = d/(\sigma_0^2\omega_\perp t) = md/(\hbar t)$ is inverse fringe spacing, and $A(z, t)$ and $C(z)$ are interference peaks amplitudes and contrasts respectively. In experiments, the relative phase $\phi_-(z)$ is obtained by fitting the interference image to Eq. (4), and so we refer to Eq. (4) as the *'transversal fit formula'*. The superscript $\perp$ means we have ignored longitudinal dynamics by substituting $G(z - z', t) \approx \delta(z - z')$ in Eq. (2). In addition, the formula also assumes $\omega_\perp t \gg 1$ and $d \ll \sigma_t$ such that the overlapping transverse Gaussian can be approximated as a single Gaussian centred at the origin. Although they can be relaxed, here we consider identical mean density $n_1(z) = n_2(z) = n_0(z)$ and ignore density fluctuation $\delta n_{1,2} \ll n_0$.

This work explores the impact of longitudinal expansion on the accuracy of relative phase extraction. In other words, we go beyond Eq. (4) by including longitudinal dynamics in our analysis, where the final density after expansion and interference is written as [24]

$$\rho_{\text{TOF}}(x, z, t) = Ae^{-x^2/\sigma_t^2}\left|\int_{-L/2}^{L/2} dz' \, G(z - z', t)\sqrt{n_0(z')}e^{i\phi_+(z')/2}\cos\left(\frac{kx + \phi_-(z')}{2}\right)\right|^2, \tag{5}$$

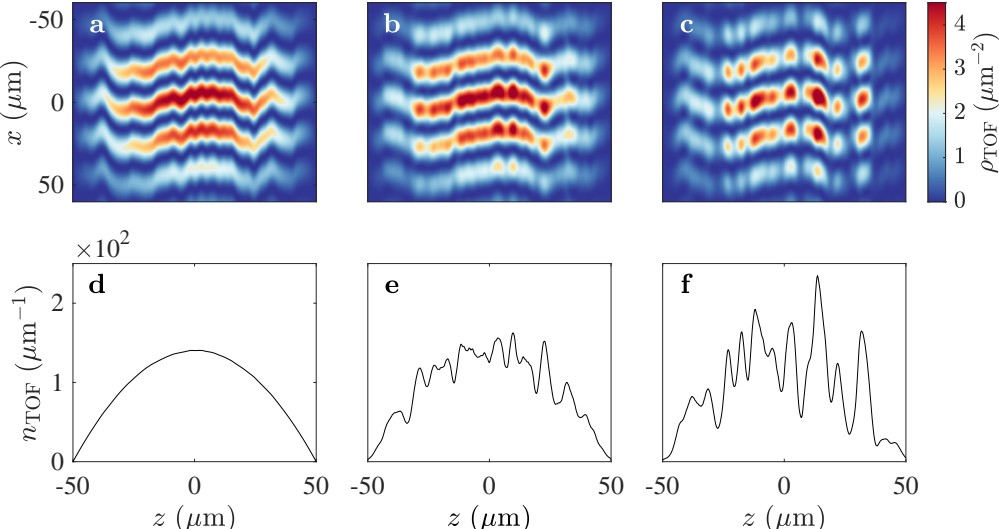

Figure 2: Comparison between three different TOF expansion models: (**a**) $\rho_{\text{TOF}}^{\perp}$, (**b**) $\rho_{\text{TOF}}$ with $\phi_+(z) = 0$, and (**c**) $\rho_{\text{TOF}}$ with $\phi_+(z) \neq 0$. Panels **d** - **f** show the respective TOF longitudinal density $n_{\text{TOF}} = \int \rho_{\text{TOF}} \, dx$. The mean insitu density $n_0(z)$ is set to follow the Thomas-Fermi approximation in harmonic potential (inverse parabola) with peak density 75 $\mu\text{m}^{-1}$. The other parameter values are $t = 15$ ms, $\omega_{\perp} = 2\pi \times 2$ kHz, $L = 100$ $\mu$m, $d = 3$ $\mu m$, and $m$ is the mass of $^{87}$Rb. These parameters are fixed throughout the paper unless stated otherwise.

where $\phi_+(z) := \phi_1(z) + \phi_2(z)$ is the common (symmetric) phase, typically unmeasured in experiments. We provide a detailed derivation of Eq. (5) in Appendix A and we show how to recover Eq. (4) from Eq. (5) in Appendix B. The mixing with common degrees of freedom in Eq. (5) is a new phenomenon neglected in Eq. (4). Meanwhile, longitudinal expansion manifests itself through the Green's function kernel which allows local correlation between density at $z$ and $z' \neq z$. We refer to Eq. (5) as the *'full expansion formula'*. Unlike the transversal fit formula, the integral form and the dependence on the common phase make it difficult to use the full expansion formula as a fit function.

We conclude our description of these models by illustrating their differences in Fig. 2a-c, showing a comparison between interference patterns of identical phase profiles computed with different expansion models. Their differences are visible through the longitudinal variation of the central peaks. They can also be seen more clearly by numerically evaluating longitudinal density $n_{\text{TOF}}(z, t) = \int dx \, \rho_{\text{TOF}}(x, z, t)$, which is directly measurable in experiments by imaging the atoms along the $x$-axis [16, 17, 22]. The result is shown in Figs. 2d-e with the transverse fit formula showing no density ripples [Fig. 2d], i.e. $n_{\text{TOF}}(z) = n_0(z)$, in contrast with the full formula [Figs. 2e-f].

The density ripples imply the presence of *systematic* longitudinal correlations in the interference pattern induced by free expansion, which is neglected in the transversal expansion model. Since we read out the relative phase from the interference pattern, it is natural to ask whether this density correlation will cause a systematic correlation in the readout phase as well, leading to a systematic error between *true* insitu phase and the readout phase. This error is indeed numerically reported in Ref. [24] but with no systematic characterization of their behaviour. We will discuss this in the next section.

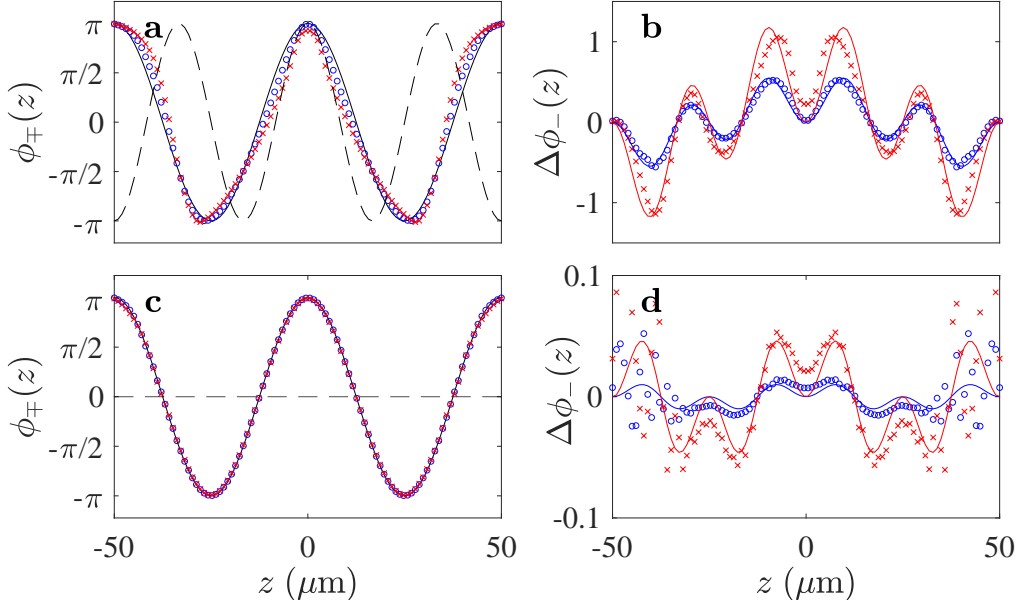

Figure 3: Simple model illustrating the different contributions of relative and common phase on the measured interference patterns in time-of-flight. (**a**) Input relative (solid black line) and common (dashed black line) phase profiles $\phi_-(z) = \pi \cos(4\pi z/L)$ and $\phi_+(z) = \pi \cos(6\pi z/L)$ together with the extracted phase profiles $\phi_-^{(\text{out})}(z)$ with $t = 7$ ms (blue circles) and $t = 15$ ms (red crosses). (**b**) Phase shift induced by longitudinal expansion $\Delta\phi_-(z,t) = \phi_-(z) - \phi_-^{(\text{out})}(z)$. The solid lines are analytical curves calculated with Eq. (9). Panels **c-d** are repetition of **a-b** with $\phi_+(z) = 0$. The initial mean density profile is the same as in Fig. 2 (inverse parabola).

# 3 Readout phase error due to longitudinal expansion

In experimental analysis, longitudinal dynamics are often ignored, and Eq. (4) is used to read out the relative phase from the density interference pattern. If we relax this assumption, the expression for the final density is given by Eq. (5), which is considerably more complicated and difficult to use as a fitting function. Our aim in this section is to assess the modelling error that may arise from ignoring longitudinal expansion. We do this by treating the integral in Eq. (5) *perturbatively*.

We start by defining an integrand function,

$$I(x,z',t) \equiv \sqrt{n_0(z')} e^{i\phi_+(z')/2} \cos\left(\frac{kx + \phi_-(z')}{2}\right), \tag{6}$$

so that the integral in Eq. (5) can be written as $\int_{-L/2}^{L/2} dz' \, G(z-z',t) I(x,z',t)$. Similar to the stationary phase approximation, the integrand's dominant contribution will come from $z' \approx z$. We may then perform asymptotic expansion of the integral around that point in analogy to Laplace's method [30], i.e. we perform Taylor expansion of $I(x,z',t)$ centred around $z$.

We show in Appendix C that Eq. (5) can always be expressed in the following form

$$\rho_{\text{TOF}}(x,z,t) \approx A'(z,t) e^{-x^2/\sigma^2(t)} \left[1 + C'(z,t) \cos(kx + \phi_-(z) - \Delta\phi_-(z,t))\right], \tag{7}$$

where $A'(z,t)$, $C'(z,t)$ now include corrections from longitudinal expansion. The form of Eq. (7) is expected from Eq. (3). However, using our integrand expansion technique, we are

able to express the fit parameters in terms of in-situ field variables. In particular, we find that longitudinal expansion introduces a *systematic phase shift* $\Delta\phi_-(z,t)$ into the readout phase, so that

$$\phi_-^{(\text{out})}(z,t) = \phi_-(z) - \Delta\phi_-(z,t). \tag{8}$$

For gases with slowly-varying mean densities, the dominant corrections for the phase $\Delta\phi_-(z,t)$ are expressed in terms of scaled derivatives of the phases

$$\Delta\phi_-(z,t) \approx (\partial_\eta\phi_-)(\partial_\eta\phi_+) - \frac{1}{2}(\partial_\eta^2\phi_-)(\partial_\eta\phi_-)^2 + O(\partial_\eta^4), \tag{9}$$

where derivatives are taken with respect to a scaled coordinate $\eta = z/\ell_t$ with $\ell_t = \sqrt{\hbar t/(2m)}$ being the length scale of longitudinal expansion. In the standard Bogoliubov theory for 1D gas [31], the scaled derivative of the phase with respect to a finite lattice length is considered a small parameter. Similarly, our formula is expanded with respect to small parameters $\partial_\eta\phi_\pm$ with $\ell_t$ being analogous to lattice length. Consequently, our formula is only valid to describe fluctuations with momenta $q < \ell_t^{-1}$. The corrections to Eq. (9) are of order four or higher in scaled phase derivatives. We note that the form in Eq. (9) already uses a linearization of an arctan function. When considering modes close to $q \sim \ell_t^{-1}$, one might need to adopt the full analytical form derived in Appendix C.

Equations (7)-(9) are the main analytical results of this paper. They can be useful to assess the reliability of the existing phase readout protocol. For example, Eq. (9) shows that the readout error grows with a longer expansion time. This is intuitive since a longer longitudinal expansion time would lead to a more systematic longitudinal correlation spread along the gas. Moreover, Eq. (9) clearly shows a dominant phase shift correction due to mixing with the common phase, which was previously unnoticed. We also find a higher-order correction term that depends only on the derivatives of the relative phase, signifying a systematic error purely due to the presence of longitudinal Green's function.

We compare our analytical prediction with numerical data by encoding smooth phase profiles, e.g. $\phi_-(z) = \pi\cos(4\pi z/L)$ and $\phi_+(z) = \pi\cos(6\pi z/L)$, into density interference pattern computed with the *full expansion formula* and then decode the relative phase with the *transverse fit formula*. We find agreement between numerical data and our analytical prediction up to finite size effects near the boundary [Fig. 3]. We also examined various other smooth profiles and obtained similar results. Note that the numerical data does not assume uniform density and yet Eq. (9) fits the data quite well, demonstrating the usefulness of our formula in realistic scenarios where mean density varies sufficiently slowly.

## 4 Reconstruction of physical quantities

Ultimately, we are interested in reconstructing physical quantities associated with the gases' initial state, which we assume to be given by a thermal state of the sine-Gordon Hamiltonian [32]

$$H = H_{TLL}(\delta n_+, \phi_+) + H_{TLL}(\delta n_-, \phi_-) - 2\hbar J n_0 \int dz \, \cos\phi_-(z), \tag{10}$$

where $H_{TLL}$ is the Tomonaga-Luttinger liquid Hamiltonian [33] in terms of the symmetric and antisymmetric density fields $\delta n_\pm = \delta n_1(z) \pm \delta n_2(z)$ and phase fields $\phi_\pm(z) = \phi_1(z) \pm \phi_2(z)$

$$H_{TLL}(\delta n_\pm, \phi_\pm) = \int dz \left[ \frac{\hbar^2 n_0(z)}{4m}(\partial_z\phi_\pm(z))^2 + g_{1D}(\delta n_\pm(z))^2 \right], \tag{11}$$

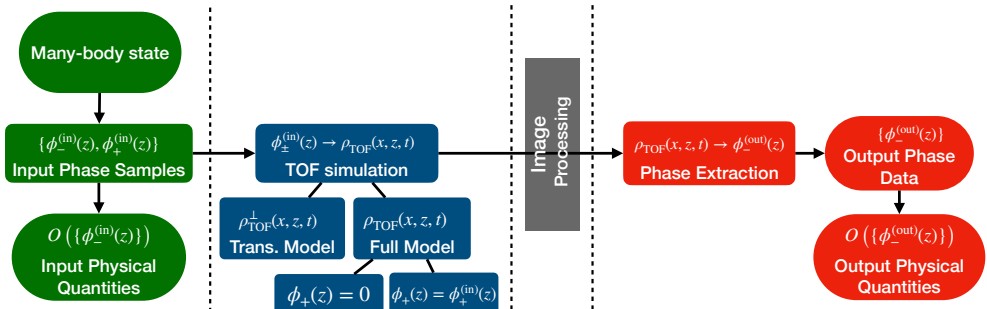

Figure 4: The simulation workflow is divided into four stages, separated by the dotted lines. The first stage (green boxes) represents the input to the simulation, obtained by sampling relative and common phase profiles from an input state. The next stage (blue boxes) represents TOF encoding implemented with three different models. The last stage (red boxes) represents decoding where relative phases and physical quantities are inferred by fitting with the transverse fit formula (4). The additional image processing stage between the encoding and decoding process simulates experimental imaging effects. We will momentarily ignore this in Sec. 4 and revisit it in Sec. 5. The goal of the simulation is to compare the input and output physical quantities.

with $g_{1D}$ being the interatomic repulsion strength in the 1D regime [27]. While the common mode is determined by this Gaussian theory, the non-Gaussianity of the relative degrees of freedom can be experimentally tuned via the single particle tunnelling strength $J$, giving rise to the sine-Gordon model. The relevance of the cosine potential can be characterized by $\chi = \lambda_T / l_J$ which is directly related to the experimentally accessible coherence factor $\langle \cos(\phi_-) \rangle$. The thermal coherence length $\lambda_T = \hbar^2 n_{1D} / (m k_B T)$ for uniform gas $n_0(z) = n_{1D}$ and phase locking length $l_J = \frac{1}{2} \sqrt{\hbar/mJ}$ determine the randomization and restoration of the phase due to temperature and tunnel coupling respectively. In thermal equilibrium phase, correlation functions for various $\chi$ (implemented by varying the tunnel coupling strength $J$) have been experimentally computed up to the $10^{\text{th}}$ order [20] and found in agreement with the predictions of the sine-Gordon model.

In this section, we assess the reliability of TOF measurement for such a task, especially in the light of possible error propagation from $\Delta\phi_-(z,t)$. We begin with investigating the reconstruction of physical quantities associated with uncoupled Luttinger liquid ($J = 0$) in thermal equilibrium in Subsecs. 4.1-4.4. We then discuss the TOF reconstruction of second-order and fourth-order correlations of the coupled sine-Gordon theory in the Gaussian ($q = 0.5$) and non-Gaussian regime ($q = 3$) in Subsec. 4.5. From here onwards, we mainly resort to numerical simulation, where our workflow is summarized in Fig. 4. The code used to perform the simulation is available in Ref. [34].

- **Independent sampling of relative and common phase profiles.** We sample many instances of $\{\phi_{\mp}^{(in)}(z)\}$ from a many body state. In our case, the many body state would either be a thermal Gaussian state or a non-Gaussian sine-Gordon state. The phase profiles corresponding to thermal Gaussian state are sampled from a multivariate normal distribution following a thermal covariance matrix [35], with small tunnelling $J = 0.1$ Hz to renormalize the zero modes. Meanwhile, the non-Gaussian phase profiles are sampled by a stochastic process described by an Itô equation [36, 37].

The sampled phase profiles are the input to our simulation. Using these inputs, the ground-truth physical quantities $\mathcal{O}\left(\left\{\phi_-^{(in)}(z)\right\}\right)$ can be computed. Although it may contain statistical fluctuations, given sufficiently many samples of the phase profiles, the computed quantities $\mathcal{O}$ should closely match their theoretical values.

- **Simulation of the TOF encoding of phases into density interference patterns.** Given the phase profiles, we simulate TOF by computing density after TOF ($\rho_{\text{TOF}}$) using Eq. (5) with varying expansion time $t$. To control for the influence of common phases, we perform the simulation twice for every $t$, once with zero common phase ($\phi_+(z) = 0$) and the second time with the sampled common phase ($\phi_+(z) = \phi_+^{(\text{in})}(z)$). In addition, we simulate the transverse expansion model to control for numerical error in the relative phase decoding process (explained below).

- **Decoding the relative phase from interference patterns.** With the obtained $\rho_{\text{TOF}}$, we use Eq. (4) as a fitting function to extract $\phi_-(z)$. To do so, we solve a constrained optimization $\phi_-^{(\text{out})}(z) \in [-2\pi, 2\pi]$ problem using the interior-point algorithm. We initialize the optimizer by feeding a linear function $\phi_-^{(0)}(z) = -kx_{\text{max}}$ where $x_{\text{max}}$ is the transversal peak position at fixed $z$ [Appendix D]. Due to phase multiplicity over a $2\pi$ period, we sometimes observe phase jumps (discontinuity) in the optimization output. We eliminate the discontinuity by applying a *phase unwrapping* protocol: adding a multiple of $2\pi$ to the phase whenever detecting a jump larger than $\pi$ until discontinuity is eliminated. However, this protocol is inaccurate for highly fluctuating profiles in finite resolution, which puts a limit on the temperatures for which our method performs reliably.

After obtaining all the decoded phases data $\{\phi_-^{(\text{out})}(z)\}$, we compute the inferred physical quantities $\mathcal{O}(\{\phi_-^{(\text{out})}(z)\})$ and compare them to the input $\mathcal{O}(\{\phi_-^{(\text{in})}(z)\})$ in different scenarios. Note that in Fig. 4, there is an additional image processing stage between the encoding and decoding process. This is the stage where the initial interference pattern gets modified due to the experimental setup and limitations of the imaging devices. We will momentarily ignore this stage and revisit it in Sec. 5.

## 4.1 Vertex correlation function

We first consider the reconstruction of vertex correlation function

$$C_\phi(z, z') = \left\langle \cos\left[\phi_-(z) - \phi_-(z')\right] \right\rangle, \tag{12}$$

where $\langle . \rangle$ denotes average over realizations. This quantity can be evaluated analytically [33] for Gaussian theory with quadratic Hamiltonian. For uniform 1D Bose gases in thermal equilibrium, we expect exponential decay of correlation with a length scale of thermal coherence length $\lambda_{T_-} = \hbar^2 n_0/(m k_B T_-)$ with $T_-$ being the temperature of the relative phase. This quantity is also useful for probing out-of-equilibrium experiments, e.g. observing light cone emergence of thermal correlation [10] and recurrences [6] in parallel 1D Bose gases.

Here, we only consider a middle cut $C_\phi(z, 0)$. The comparisons between input and reconstructed $C_\phi(z, 0)$ for different parameters are shown in Fig. 5. We find that the reconstruction of phase correlation function $C_\phi$ is robust against systematic phase shift due to longitudinal dynamics during TOF, i.e. $\Delta\phi_-$ does not influence the reconstruction of $C_\phi$. This is intuitive since $C_\phi$ mostly depends on the low mode and long wavelength fluctuations which have small derivatives and thus do not get significantly influenced by $\Delta\phi_-$.

## 4.2 Full distribution function

Shot-to-shot variations of the interference between two parallel 1D Bose quasicondensates can reveal signatures of quantum fluctuation [19, 38]. A key question is how much of the observed fluctuations and their correlations are fundamentally quantum, especially in systems

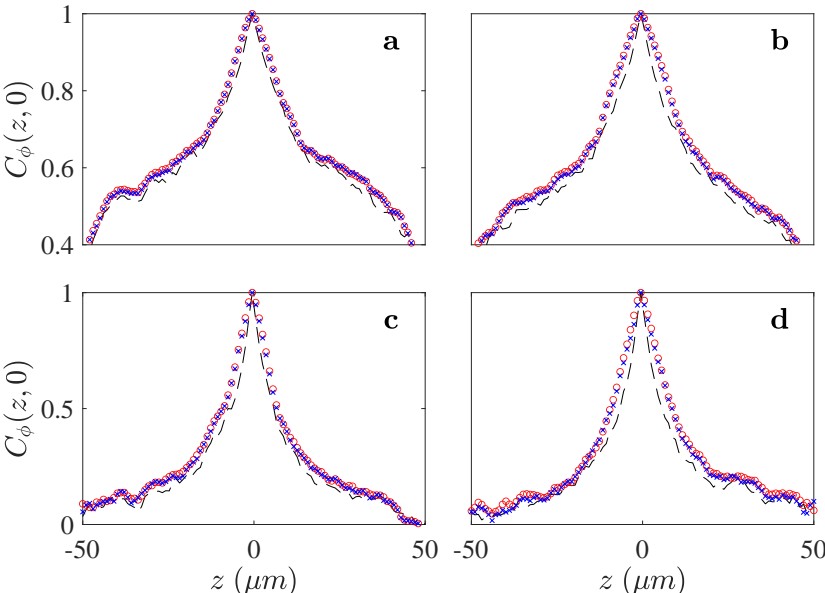

Figure 5: The input (dashed black line) and output vertex correlation function $C_\phi(z, 0)$ for $T_- = 25$ nK (**a-b**) and $T_- = 75$ nK (**c** - **d**) reconstructed with 500 TOF simulations. The first (**a**, **c**) and second (**b**, **d**) columns are reconstructed with 7 ms and 15 ms, respectively. The blue crosses (red circles) are data from TOF simulation with $\phi_+(z) \neq 0$ ($\phi_+(z) = 0$) sampled from the same thermal distribution as the relative phase ($T_+ = T_-$).

with finite temperatures. Here, we will probe the full distribution function of the interference contrast $P(\xi)$ defined by

$$\xi(l) = \frac{\left| \int_{-l/2}^{l/2} e^{i\phi_-(z)}\, dz \right|^2}{\left\langle \left| \int_{-l/2}^{l/2} e^{i\phi_-(z)}\, dz \right|^2 \right\rangle}, \tag{13}$$

with $l$ being a variable distance from 0 to $L$. In our case, we are probing the effect of time-of-flight on the reconstruction of $P(\xi)$ in equilibrium. In out-of-equilibrium, the full distribution function $P(\xi)$ has also been used to study pre-thermalization dynamics after coherent splitting [9, 11, 12].

We compare the input and reconstructed (output) full distribution function $P(\xi)$ for three different length scales $l$ in Fig. 6. We find that except for a minor reduction in the high-contrast probability, the qualitative features of the input and output distribution almost coincide. The suppression of the high-contrast probability implies that, as expansion time becomes longer, the medium contrast becomes over-represented and so it could slightly modify the skewness of the underlying distribution. We believe this is due to additional fluctuation coming from the systematic phase shift $\Delta\phi_-(z, t)$ which grows with expansion time. Furthermore, by comparing the first and second rows in Fig. 6, we show that the common phase does not significantly influence the full distribution function. Overall, we observe the same qualitative transition for different $l$ as reported in Refs. [19, 38]. Thus, longitudinal expansion and common phase do not play significant roles here and the existing phase readout protocol can faithfully reproduce the full distribution function $P(\xi)$.

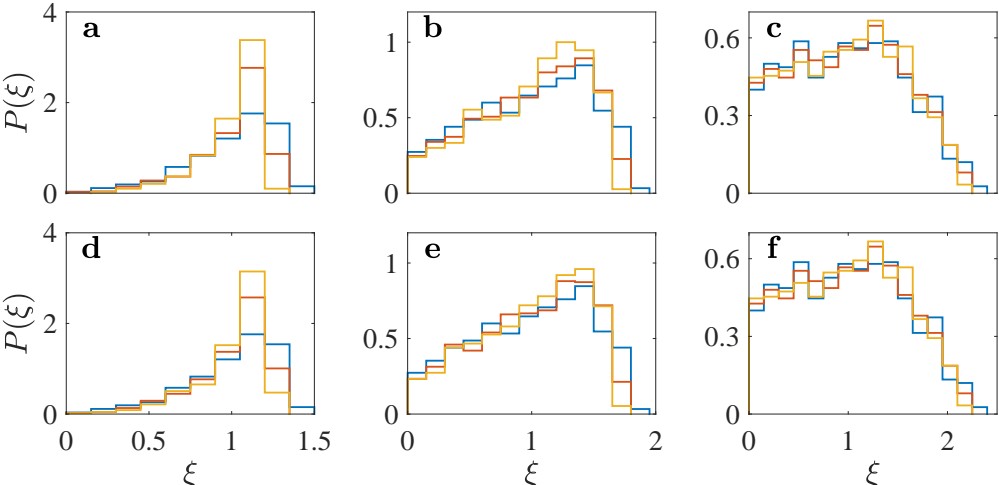

Figure 6: Full distribution function $P(\xi)$ computed with 1000 phase profiles sampled from a thermal state with $T_\pm = 75$ nK. The top (bottom) row **a-c** (**d-f**) corresponds to the case where $\phi_+(z) = 0$ ($\phi_+(z) \neq 0$). The length scales are $l = 9.8$ $\mu$m (**a,d**), $l = 25.5$ $\mu$m (**b,e**), and $l = 49$ $\mu$m (**c,f**). The blue histogram is the input, red (yellow) histogram is the reconstructed distribution from 7 ms (15 ms) full expansion.

## 4.3 Velocity-velocity correlation

The spatial derivative of phase has a physical meaning as a velocity field $u_\pm(z) = (\hbar/m)\partial_z\phi_\pm(z)$ in the hydrodynamics description of a superfluid. Here, we specifically look at the correlation in the relative velocities

$$C_u(z,z') = \langle\partial_z\phi_-(z)\partial_{z'}\phi_-(z')\rangle - \langle\partial_z\phi_-(z)\rangle\langle\partial_{z'}\phi_-(z')\rangle .\qquad(14)$$

If the relative velocities of the atoms at $z$ and $z'$ are independent, then $C_u(z,z')$ vanishes. Any non-zero values (discounting statistical fluctuation) for this quantity reflect a correlation in the relative velocities, i.e. if $C_u(z,z') > 0$ the relative velocities of the atoms at $z$ and $z'$ tend to align whereas if $C_u(z,z') < 0$ they tend to be opposite. Recently, the velocity-velocity correlation has been measured in experiments to observe curved light cones in a cold-atomic quantum field simulator [7]. We compare the input and output velocity correlation in Fig. 7. The in situ velocity correlation $C_u^{(in)}(z,z')$ for a thermal state is not completely diagonal. Instead, it has a weak and short-distance anti-correlation as shown by Fig. 7a.

Interestingly, we observe spatial propagation of the initial anti-correlation in the TOF model with longitudinal expansion shown in Figs. 7c-d and Figs. 7e-f, which does not appear in the control simulation with only transversal expansion [Fig. 7b]. We observe the length scale for this correlation (the span of the off-diagonal) increases with a longer expansion time. Such propagation of correlation can be physically understood in a quasi-particle picture, where neighbouring quasi-particles with initial opposite velocity correlation will move further away from each other as the gas expands longitudinally. We also observe alternating patterns of positive and negative correlation which indicates momentum interference in the longitudinal direction [Fig. 7e]. However, this long-distance correlation and anti-correlation are randomized when common phases are involved and only the propagation of the primary anti-correlation persists [Fig. 7f].

This propagation is similar to what has been observed experimentally in the context of a quench from an interacting to non-interacting pair of Luttinger liquids [7]. The difference here is that we report the propagation of velocity correlation due to quenching into a free Hamiltonian induced by the TOF measurement protocol. To observe this, one must resolve fluctuations

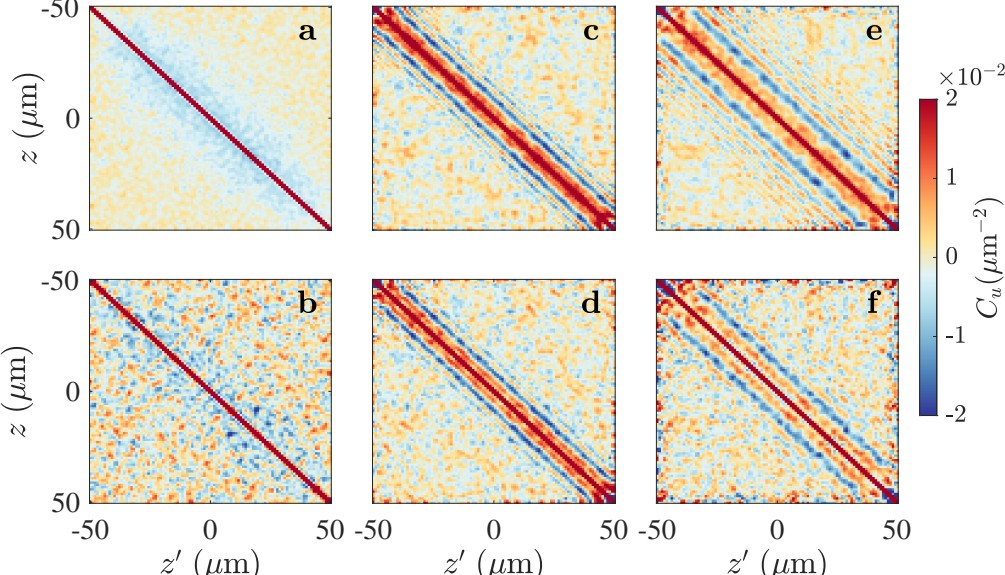

Figure 7: Velocity-velocity correlation $C_u(z, z')$ calculated from (**a**) input phase profiles and (**b**) extracted profiles with 7 ms transversal expansion (**c**) 7 ms full expansion with $\phi_+(z) = 0$ and (**d**) $\phi_+(z) \neq 0$. Panels (**e**)-(**f**) are the same as (**c**)-(**d**) except for $t = 15$ ms. Panel (**a**) is generated with $10^4$ phase profiles whereas panels (**b**)-(**f**) are generated with 500 TOF simulation. The upper bound of the color bar has been adjusted to low values to accentuate structures in the off-diagonals. The input phase profiles are sampled from a thermal distribution with temperatures $T_\pm = 75$ nK.

with a length scale comparable to the length scale of TOF dynamics $\ell_t = \sqrt{\hbar t/(2m)} \approx 2.3$ $\mu$m for $t = 15$ ms. Thus, this effect might not be captured by present experiments (see Sec. 5) and by our perturbative treatment in Eq. (9). Nevertheless, our numerical results point to the necessity of calibrating the results of dynamical propagation of velocity-velocity correlation such as in Ref. [7] to the measurement background in future experiments with enhanced resolution.

## 4.4 Mean Fourier spectrum & temperature

The mean Fourier spectrum $\langle |\Phi_q|^2 \rangle$ where $\Phi_q = (1/\sqrt{L}) \int_{-L/2}^{L/2} e^{-iqz} \phi_-(z) \, dz$ is another relevant physical quantity of the gases. Similar to the vertex correlation function (Subsec. 4.4), it also encodes information about temperature in equilibrium

$$\langle |\Phi_q|^2 \rangle = \frac{mk_B T_-}{\hbar^2 q^2 n_0} = \frac{\alpha_{T_-}}{q^2}, \tag{15}$$

where $\alpha_{T_-} = 1/\lambda_{T_-} = mk_B T_-/(\hbar^2 n_0)$ is inverse thermal coherence length. The mean Fourier spectrum $\langle |\Phi_q|^2 \rangle$ is relevant in quadrature tomography technique for extracting relative density fluctuation information from the relative phase data [35], which is then used to probe the covariance matrix of the system in and out of equilibrium [13, 39]. Here, we compare the input and output spectrum for a thermal state with a fixed temperature $T_- = 50$ nK. Then, we vary the temperature and fit $\langle |\Phi_q|^2 \rangle$ according to Eq. (15) to extract $\alpha_{T_-}$. For our simulation, we will assume the relative and common degrees of freedom are in thermal equilibrium with respect to each other ($T_+ = T_-$).

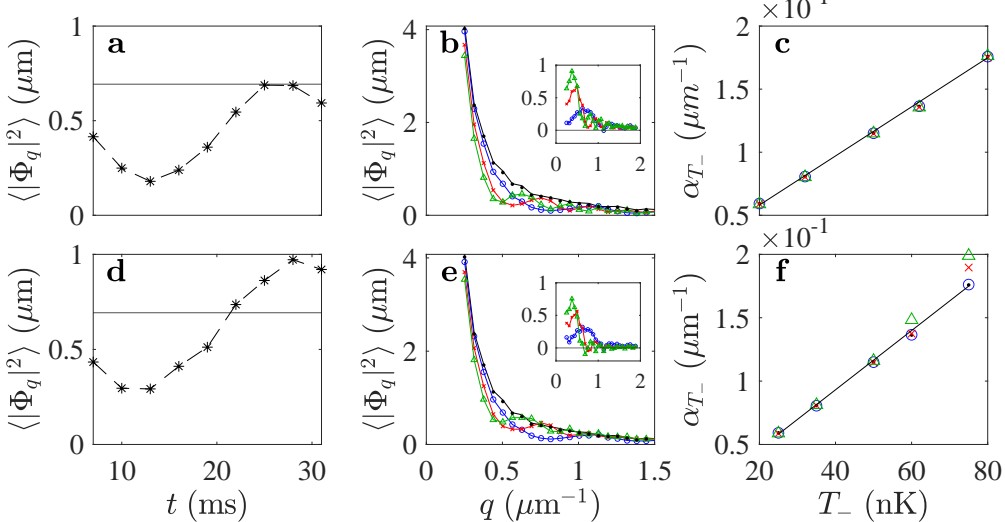

Figure 8: Thermal ($T_- = 50$ nK) mean Fourier coefficients $\langle|\Phi_q|^2\rangle$ computed using 200 realizations of TOF measurement simulation. In panel **a**, $q = 20\pi/L$ ($\approx 0.63$ $\mu$m$^{-1}$) is fixed but expansion time is varied. In panel **b**, $k$ is varied but expansion times are fixed at different values: $t = 7$ ms (blue circles), $t = 15$ ms (red crosses), and $t = 30$ ms (green triangles). The black solid lines are the ground truths computed from the input data. To emphasize the oscillation in the intermediate mode, we only plot data points with $k \geq 10\pi/L \approx 0.31$ $\mu$m$^{-1}$. The inset shows the residue $\Delta_k = \langle|\Phi_q^{(\text{in})}|^2\rangle - \langle|\Phi_q^{(\text{out})}|^2\rangle$. Panel **c** shows inverse thermal coherence length $\alpha_{T_-}$ as a function of temperature $T_-$. Panels **d-f** are the repetition of **a-c** but include common phase with temperature $T_+ = T_-$.

We start by fixing the momentum to be $q = 20\pi/L$ and vary the expansion time $t$. In the absence of energy transfer from other sectors, the energy in phase quadrature can not exceed its initial energy, which explains why the output spectrum appears to be upper-bounded by its in situ values [Figs. 8a,b]. However, for high enough common phase temperature, the in-situ values do not provide an upper bound anymore because initial common phase fluctuation can give extra energy to the relative phase [see Eq. (9)]. Note that in our case, energy from in situ density fluctuation is ignored.

While a perfectly faithful reconstruction of $\langle|\Phi_q|^2\rangle$ should not depend on expansion time $t$, we find a non-trivial oscillation of $\langle|\Phi_q^{(\text{out})}|^2\rangle$ with respect to $t$ attributed to longitudinal expansion [Figs. 8a,d]. This oscillation is also visible when we plot $\langle|\Phi_q|^2\rangle$ as a function of $q$ for different values of expansion time as shown in Figs. 8b,e. To emphasize the oscillation in the intermediate mode regime we have omitted the low-momentum population $q < 10\pi/L$. The insets in Figs. 8b,e show the residue between input and output spectrum $\Delta_q = \langle|\Phi_q^{(\text{in})}|^2\rangle - \langle|\Phi_q^{(\text{out})}|^2\rangle$, which qualitatively resembles the evolution of density ripple spectrum [16]. As expansion time gets longer, the maximum of the residue $\Delta_q^{(\text{max})} = \max_q(\Delta_q)$ grows and its peak location $q_{\text{max}}$ shifts to a lower mode. We note that this effect is due to dynamics in the high momentum modes which goes beyond the correction in Eq. (9). Indeed, for $t = 15$ ms the expansion length scale is $\ell_t = \sqrt{\hbar t/(2m)} \approx 2.3$ $\mu$m giving a momentum cutoff $q \sim \ell_t^{-1} \approx 0.43$ $\mu$m which is smaller than the typical momenta where this oscillation is observed.

We checked numerically that such oscillation originates from a transfer of energy from the relative phase to relative density fluctuation during the expansion. Note that although we ignore density fluctuation *in situ*, it does not prevent density fluctuation from developing as the cloud expands. Indeed, density ripples displayed in Figs. 2e-f can be considered as the common density fluctuation of the clouds after TOF. In contrast, the relative density fluctuation, i.e. $\delta n_-^{(\text{tof})}(z,t) = \int d^2\vec{r}\left[|\psi_1(\vec{r},z,t)|^2 - |\psi_2(\vec{r},z,t)|^2\right]$ is not directly measurable in experiments, but can still be computed in simulations. We found opposite oscillatory behaviour between the spectrum of $\delta n_-^{(\text{tof})}(z,t)$ and the lost energy in $\langle|\Phi_q^{(\text{out})}|^2\rangle$, suggesting energy transfer between the two fields.

Finally, we check the impact of this oscillation to the reading of temperature using Eq. (15). We perform fitting $\langle|\Phi_q|^2\rangle = \alpha_{T_-} q^{-2}$ for different values of $T_-$ and then plot $\alpha_{T_-}$ as a function of $T_-$ shown in Figs. 8c,f. We find that the oscillation due to longitudinal expansion does not significantly affect the readout of temperatures. However, the additional fluctuation from common phase does make a difference for medium to long expansion time ($t > 15$ ms) and high enough $T_+ \geq 60$ nK.

## 4.5 Gaussian and non-Gaussian correlation functions

Equal-time higher-order correlations contain detailed information about the many-body state and can be directly calculated from the extracted phase profiles after time-of-flight [20, 21]. Computing all correlation functions is tantamount to solving a many-body problem. The $N$-th order relative phase correlation function referenced at $z = 0$ is defined by

$$G^{(N)}(\mathbf{z}) = \left\langle \prod_{i=1}^N (\phi_-(z_i) - \phi_-(0)) \right\rangle, \tag{16}$$

where $\mathbf{z} = (z_1, z_2, ..., z_N)$. In general, the correlation function can be decomposed into the connected and disconnected part

$$G^{(N)}(\mathbf{z}) = G_{\text{con}}^{(N)}(\mathbf{z}) + G_{\text{dis}}^{(N)}(\mathbf{z}). \tag{17}$$

The disconnected part can be expressed in terms of lower-order correlations while the connected part contains genuine new information about $N$-body interactions [20, 21]. The computation of correlation function of order larger than two is analytically difficult, except for special cases such as non-interacting Gaussian states, where higher-order connected correlations vanish identically for $N > 2$.

We first compare the second order correlation $G^{(2)}(z_1, z_2)$ for sine-Gordon Hamiltonian in Gaussian ($\chi \equiv \lambda_T/l_J = 0.5$) and non-Gaussian ($\chi = 3$) regimes. The comparison is shown in Fig. 9. We observe only small differences between input and output correlation in the small $J$ Gaussian regime of the sine-Gordon model, implying that TOF can faithfully reconstruct Gaussian correlation. However, in non-Gaussian regimes, we observe a spread of cross-shaped strips at the center, which can be interpreted as a correction from higher-order correlation terms induced by systematic phase shift error.

Next, we compare the input and TOF reconstruction of fourth-order correlation function $G^{(4)}(z_1, z_2, z_3, z_4)$. The connected part $G_{\text{con}}^{(4)}(z_1, z_2, z_3, z_4)$ strictly vanishes for Gaussian states. We have checked that in the Gaussian regime of $\chi = 0.5$, the four-point correlation function indeed factorizes into the products and sum of contributions coming from the two-point function (disconnected part), barring some fluctuations due to finite statistics (see Fig. 22 in Appendix E). Here, we will focus only on the $G^{(4)}(\mathbf{z})$ reconstruction of non-Gaussian states, which contains information about four-body correlation. However, a direct comparison between input and output correlation functions is not straightforward for higher dimensional data. For visualization, we fix a cut at two different lengths $z_3 = -z_4 = 5.5$ $\mu$m and $z_3 = -z_4 = 10$ $\mu$m.

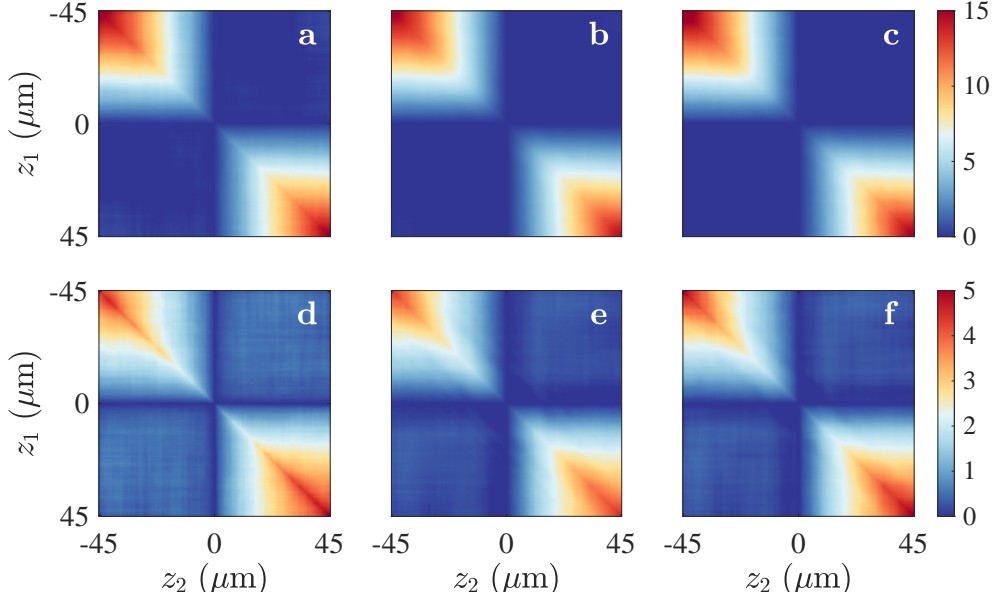

Figure 9: Second-order correlation $G^{(2)}(z_1, z_2)$ for $\chi \equiv \lambda_T / l_J = 0.5$ (**a** - **c**) and $\chi = 3$ (**d** - **f**). The first column (**a**, **d**) shows the input correlation $G^{(2)}_{\text{in}}(z_1, z_2)$, the second column (**b**, **e**) shows TOF reconstruction $G^{(2)}_{(\text{out})}(z_1, z_2)$ with $t = 15$ ms and zero common phase $\phi_+ = 0$, and the third column includes the effect of common phase sampled from a thermal distribution with $T_+ = 75$ nK. The edge data of length 2.5 $\mu$m on each end have been omitted to suppress boundary effects.

From Fig. 10, we find that in both cases TOF allows a faithful reconstruction of the connected correlator. However, the disconnected part appears to be modified by TOF as shown in Fig. 11. The effect of TOF is especially profound for short distance cut at $z_3 = -z_4 = 5.5$ $\mu m$ (Figs. 11a-c) where we find correlations which are different from insitu not only quantitatively but also qualitatively. In this regime, the correlation is dominated by systematical deviations generated by longitudinal expansion.

We hypothesize that this systematic is due to the movement of the atoms during longitudinal expansion, i.e. the atoms at $z$ insitu already move a distance of $\sim z \pm \ell_t$ after time-of-flight which is a physical mechanism behind the systematic phase shift error in Eq. (9). This can also be seen from Fig. 9 where the cut $z_3 = -z_4 = 5.5$ $\mu$m is still located in the region dominated by TOF systematics, i.e. the expanding cross region in the middle of Fig. 9. Consequently, it introduces an extra positive correlation in the off-diagonal block and a negative correlation in the diagonal block. When we probe longer distance cut $z_3 = -z_4 = 10$ $\mu$m, however, the input and reconstructed correlation appear more similar to each other than the shorter cut, but still with a slight asymmetry between the diagonal and off-diagonal blocks, and a discrepancy in the absolute value of the correlation.

Our results highlight the importance of considering measurement systematics from time-of-flight when looking into higher-order correlation data [20]. Although the connected part of the correlation appears conserved by TOF, the disconnected part is affected. This may then distort the overall result, i.e. measure of non-Gaussianity. However, this systematic effect is dominant in short-length scale of $\sim 5$ $\mu$m as compared to the typical cut in experiments of around $\sim 15$ $\mu$m. A shorter cut is usually not taken in experiments due to the blurring of imaging systematics, which will be explained in the next section.

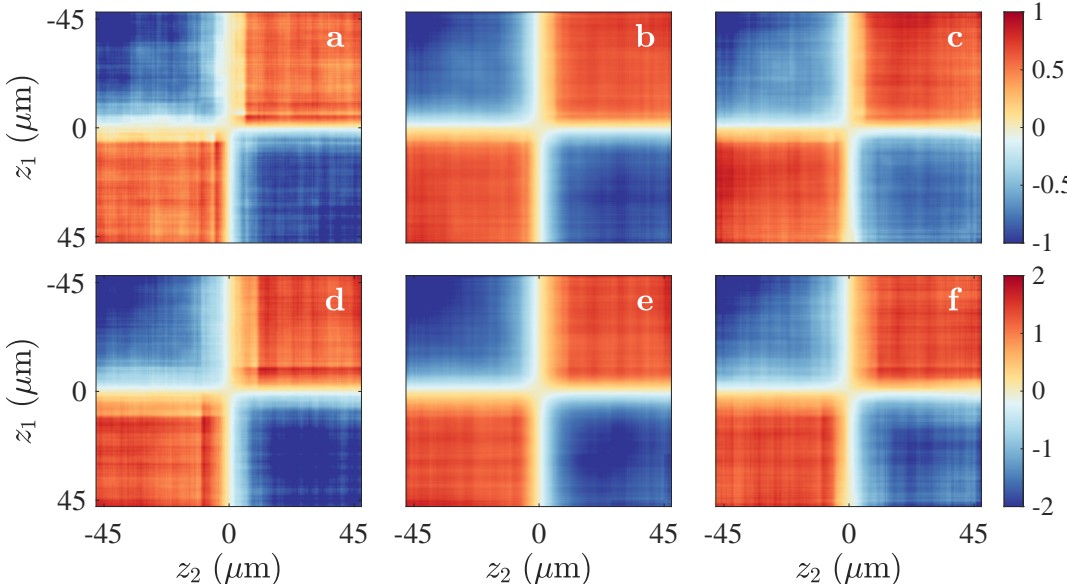

Figure 10: Time-of-flight (TOF) reconstruction of connected fourth-order correlation function $G_{con}^{(4)}$ cut at $z_3 = -z_4 = 5.5\ \mu m$ (**a**-**c**) and $z_3 = -z_4 = 10\ \mu m$ (**d**-**f**) for $\chi = \lambda_T/l_J = 3$. All panels are reconstructed with 2500 realizations of 15 ms TOF with different expansion models. The first column (**a**, **d**) involves only transversal expansion, the second column (**b**, **e**) includes longitudinal expansion but with common phase kept at zero while the third column (**c**, **f**) includes both longitudinal expansion and common phase sampled from a thermal distribution with $T_+ = 75$ nK. The edge data of length 2.5 $\mu$m on each end have been omitted to suppress boundary effects.

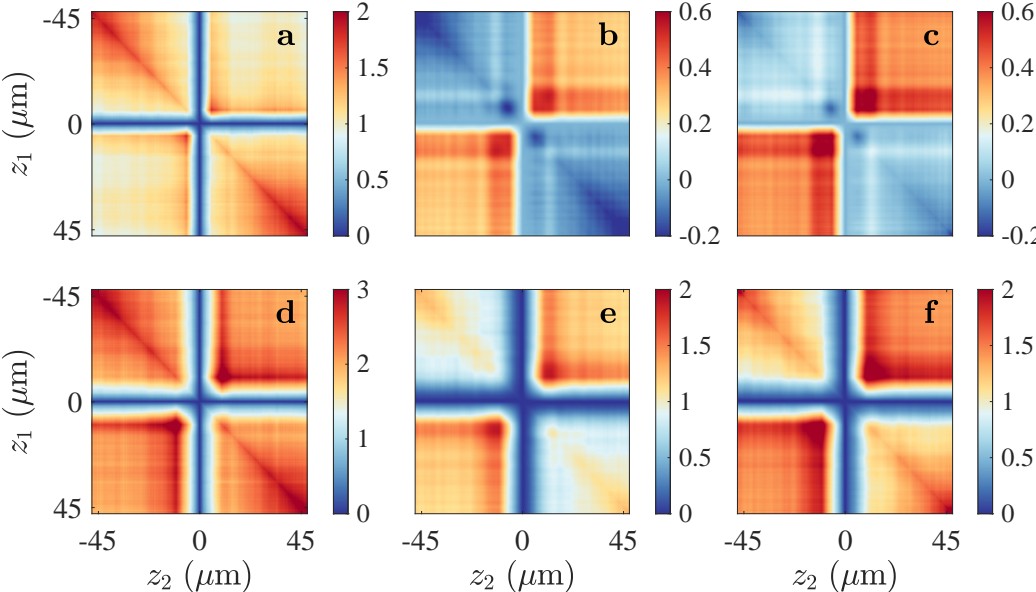

Figure 11: Time-of-flight (TOF) reconstruction of disconnected fourth-order correlation function $G_{dis}^{(4)}$ cut at $z_3 = -z_4 = 5.5\ \mu m$ (**a**-**c**) and $z_3 = -z_4 = 10\ \mu m$ (**d** - **f**) for $\chi = \lambda_T/l_J = 3$. Each panel corresponds to the same TOF models and parameter regime as in Fig. 10.

## 5 The effects of imaging

In the previous section, we discussed how the systematic error generated during longitudinal expansion propagates into the measurement of the physical properties of the gas. We find that TOF reconstruction is robust against the systematic phase shift induced by longitudinal dynamics for quantities that mostly rely on long wavelengths fluctuations such as vertex correlation function, full distribution function, and second-order correlation function. On the other hand, for some quantities such as velocity-velocity correlation, mean Fourier coefficients, and fourth-order correlation functions, the details of fluctuations might matter and so we observe some qualitative differences between the input and the reconstructed quantity.

To check if our analysis also holds in a realistic experimental setting, it is necessary to include the effects introduced by the experimental implementation of the imaging. These include foremost the finite imaging resolution of the imaging optics, the finite pixel size, and the readout noise of the camera. In an ideal setting, the readout noise is given by the photon shot noise of the detected light. In a realistic scenario, the readout noise must also include physical processes that occur during imaging. For example, as the atomic cloud scatters light, it receives a momentum transfer which can lead to diffusion of the atoms in the imaging plane. Moreover, in absorption imaging, the incoming light is in the imaging direction, which may push the image out of focus. In addition, the cloud is also falling under gravity during the exposure time. All together, they result in a washing out of short-distance patterns and lead to an effective high-frequency cut-off in the imaging function. For a detailed analysis of short wavelength (high momentum) physics, this high-frequency cut-off needs to be determined with exceptional care, mostly from numerical modelling of all the physical effects participating in the specific implementation of the measurement, see Ref. [22] for a more detailed discussion of experimental imaging systematics.

In our numerical study, we take into account these effects by processing our TOF density image with a code that simulates a realistic imaging process, fine-tuned to specific experiments as in Refs. [17, 22], e.g. the pixel size is set to be approximately 2 $\mu$m, and the defocusing of the camera is set to be 32.7 $\mu$m which consists of 25 $\mu$m recoil and 7.7 $\mu$m due to free-falling during 50 $\mu$s exposure time. Prior analysis [22] has shown that the effective result of all the imaging systematics is to induce an exponential momentum cutoff $\sim \exp(-k^2 \sigma_{\mathrm{cutoff}}^2)$ where $\sigma_{\mathrm{cutoff}}$ depends on the specifics of the experiments and system parameters. In our simulation, the cutoff is approximately $\sigma_{\mathrm{cutoff}} \sim 2.5 \,\mu$m.

The comparisons between density images before and after image processing for various expansion times are shown in Fig. 12. For a very short expansion time ($t = 1.5$ ms), the fringe spacing ($\lambda = ht/md \approx 2.3 \,\mu$m) is still too small to be resolved by the imaging. By $t = 3.5$ ms ($\lambda \approx 5.4 \,\mu$m), the interference fringe is finally resolved and one can start extracting the phase reliably, although with a significantly lower contrast as compared to the unprocessed image (see Fig. 16 in Appendix E). After $t = 3.5$ ms, the qualitative differences between the density image mostly appear in the density ripple as one can see by comparing Fig. 12g and Fig. 12h. After this limit, imaging effects only modifies the Gaussian width of the cloud and introduces momentum cutoff, thus effectively smoothening short wavelength fluctuations in the extracted fit parameters (see Fig. 13).

To check the robustness of our analytical systematic phase shift formula (9), we perform the same numerical experiment as in Sec. 3 where we encode and decode a smooth single-mode phase profiles with TOF simulation but additionally include image processing to the encoding step. The result is shown in Fig. 17 of Appendix E. We find that the dominant correction due to mixing with the common phase is still present even after taking into account imaging systematics. On the contrary, the higher order correction term that arises purely due to the Green's function gets blurred by noise and other imaging systematics. Thus, we expect that

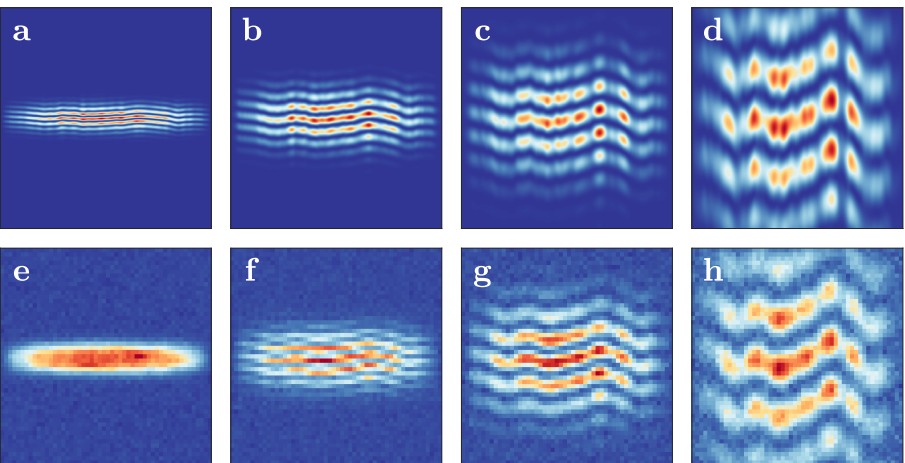

Figure 12: Time-of-flight density interference image before (**a-d**) and after (**e** - **h**) taking into account imaging effects. The expansion times are 1.5 ms (**a**, **e**), 3.5 ms (**b**, **f**) 7 ms (**c**, **g**), and 15 ms (**d,h**). The input relative and common phase profiles are identical to that of Fig. 2.

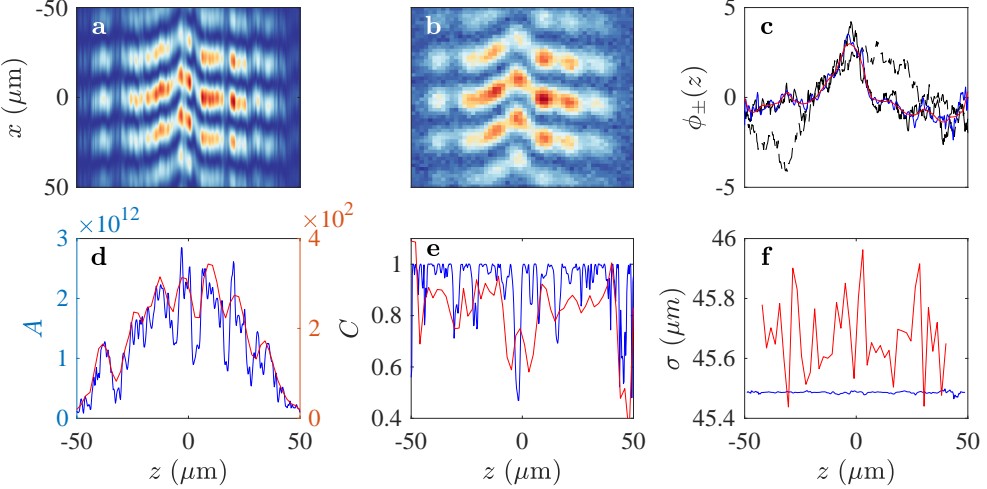

Figure 13: Comparison between single-shot relative phase extraction with and without imaging effects. Panel **a** (**b**) shows density without (with) imaging effects after 15 ms TOF. The extracted relative phase is shown in panel **c** where blue (red) denotes the result extracted from TOF density without (with) imaging effects. The black solid line is the input relative phase and the dashed black line is the common phase. Panels (**d-f**) show the other fit parameters: amplitude $A(z)$, contrast $C(z)$, and width $\sigma_t(z)$ with the blue (red) color denoting the fit parameters extracted from TOF without (with) imaging effects. This figure demonstrates that imaging only modifies the Gaussian width of the cloud and smoothens short wavelength fluctuations in the extracted fit parameters.

some features of the physical quantities that arise due to the high momentum modes dynamics will also get blurred after taking into account imaging. This is indeed what we observed in the numerical simulation for the reconstruction of physical quantities associated with Gaussian Luttinger liquid theories. For quantities that rely on long wavelength fluctuation such as vertex correlation function (4.1) and full distribution function (4.2), TOF reconstruction is insensitive to image processing (Figs. 18-19 in Appendix E). On the other hand, the qualitative effects we observe in the TOF reconstruction of Fourier spectrum and velocity-velocity correlation due to the dynamics of high mode get washed out after taking into account imaging effects (Figs. 20 -21 in Appendix E). Lastly, for the fourth-order correlation in the non-Gaussian regime, we find that the slight asymmetry between the diagonal and off-diagonal plateaus of the cut disconnected correlation is still present, although much weaker than without imaging effects (Fig. 14).

## 6 Summary & discussion

In summary, we derived an analytical expression for systematic phase shift error due to longitudinal expansion, specifically due to mixing with the common degrees of freedom and the presence of longitudinal Green's function. We also assessed the error propagation in the reconstruction of physical quantities related to the statistics of the relative phase field. We find that Gaussian observables (vertex correlation function, two-point correlation function, full distribution function) are well-preserved by time-of-flight. However, for higher moments and observables sensitive to high-momentum fluctuations, e.g. mean Fourier spectrum, velocity-velocity correlation, and fourth-order correlation functions, deviations arise due to longitudinal dynamics and so they must be taken with great care. In the case of mean Fourier coefficients and velocity-velocity correlation, the deviations are mostly due to dynamics in the high momentum modes, which lie beyond our perturbative analytical treatment and current experimental imaging resolution. However, these deviations might still be important to consider in future experiments with improved resolution. For the fourth-order correlation functions, local deviation persists even after taking into account current experimental imaging resolution. Whether our perturbative correction formula can be used to correct this deviation and to what extent this affects the measure of non-Gaussianity [20] will be explored in future work.

To improve on the readout of these 1D quantum simulators one can implement atom optical elements during the time-of-flight. Implementing a weak cylindrical lens (harmonic potential along the longitudinal direction applied for a finite time) projects the image to infinity and will result in transforming the time-of-flight measurement into a measurement of longitudinal momentum [40]. Implementing a stronger cylindrical lens leads to direct imaging of the longitudinal in-situ coordinate into the imaging plane [41]. The latter will also allow us to enlarge the image and gain considerable improvement in longitudinal imaging resolution, and it will solve problems caused by the longitudinal expansion and mixing between common and relative degrees of freedom caused by the finite temperature of the trapped quantum gas. Both options will be explored in detail in future work.

The analysis done in this paper is subjected to the validity of the modelling approximations [see Fig. 1]. One approximation we made was to ignore the broadening due to atomic repulsion $\sigma_0^2(z) = \sigma_0^2 \sqrt{1 + 2a_s n_0(z)} \approx \sigma_0^2$. Relaxing this assumption makes it difficult to obtain an analytical relation between the initial state and the final measured density due to the non-separability of the initial state. However, assuming that the non-separability is weak, there exists an ansatz [22] that can phenomenologically capture the most relevant features of interference image broadened by scattering. The ansatz is to replace all $\sigma_0$ appearing in Eq. (4) by the broadened $\sigma_0(z)$. Note that the fringe spacing now also depends on $z$, i.e.

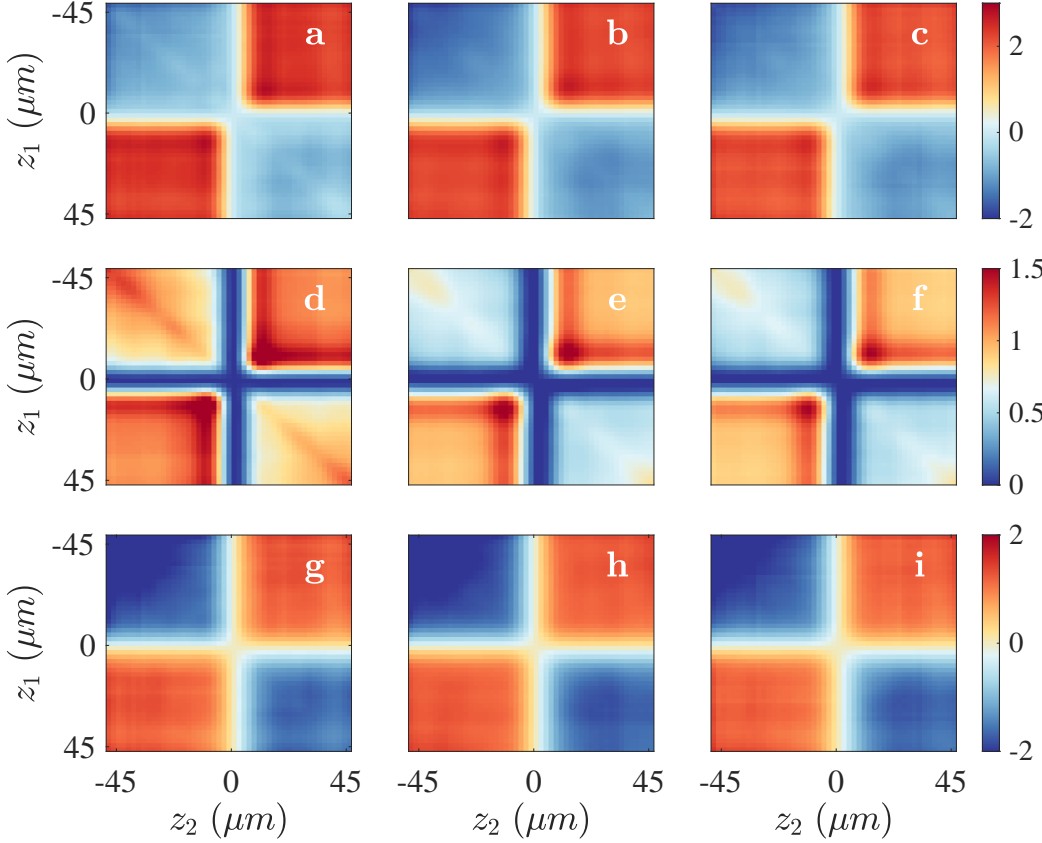

Figure 14: Time-of-flight (TOF) reconstruction of fourth-order correlation function $G^{(4)}(z_1, z_2, z_3, z_4)$ of the sine-Gordon model in the non-Gaussian regime ($\chi = \lambda_T/l_J = 3$) taking into account imaging effects. The data is cut at $z_3 = -z_4 = 11.5\ \mu m$ for visualization. Panels (**a-c**) show the full correlation, panels (**d - f**) show the disconnected part, and panels (**g - i**) shows the connected part. The first column (**a, d, g**) represents the case with only transversal expansion and imaging, the second column (**b, e, h**) includes longitudinal expansion and imaging but with common phase kept at zero and the last column (**c, f, i**) corresponds to the case with longitudinal expansion, imaging effects, and common phase sampled from thermal distribution with $T_+ = 75$ nK. Each panel is reconstructed from 2500 TOF realizations with 15 ms expansion time. The edge data of length 2.5 $\mu$m on each end have been omitted to suppress boundary effects.

$k(z, t) = d/(\sigma_0^2(z)\omega_\perp t)$. Taking longitudinal expansion into account, we can develop a similar ansatz to modify Eq. (5). We replace all $\sigma_0$ with $\sigma_0(z')$ and propagate it with a Green's function. Preliminary numerical simulation with this ansatz has revealed that scattering only affects the width of the final image, but it does not significantly affect other extracted fit parameters [Fig. 15 in Appendix E].

Furthermore, throughout this paper, we have ignored the impact of density fluctuations by assuming $\delta n_{1,2} \ll n_0$ which might not be accurate in higher temperatures. We will address the effect of density fluctuation in future work. Moreover, we have assumed that the time-of-flight expansion is fully ballistic. A more refined modelling would be to include the hydrodynamic effect at the initial phase of the expansion, where interaction energy still remains in the system. Only after interaction energy sufficiently decays, does the system follow fully ballistic

dynamics. A further direction of work will be to include the final state interaction during the initial expansion. This will be important when studying systems in the 1D-3D cross-over when the fast switch-off of interactions can no longer be guaranteed.

In addition to refining the model, another future direction is to extract the common phase from TOF interference pattern. From this study, we find that information about the common phase is imprinted on the density ripple. Density ripple has been used for thermometry in the case of single condensate [16, 17]. However, the significance of density ripple in the two condensates case has not been explored. Developing a readout method of the common phase from density ripple could be useful in unlocking the full potential of 1D Bose gas interference experiments, especially in non-equilibrium. For example, it is known that the higher order correction to the sine-Gordon model for describing tunnel-coupled 1D Bose gas involves a coupling between relative and common phase [32, 42]. Moreover, density imbalances between atoms in the two double wells can also couple the relative and common phases, leading to a double-light cone relaxation [43]. Finally, having access to a common phase might allow us to simulate spin-charge transport in 1D Bose gases [12]. This work serves as a fundamental starting point for further research in this direction.

In conclusion, our study underscores two significant findings. Firstly, it provides a comprehensive understanding of various systematics sources in local relative phase reconstruction with time-of-flight measurement. Secondly, it identifies avenues and regimes for enhancing modelling methods to achieve more accurate reconstructions. In addition, we also observe the potential for extracting additional information from TOF measurements [44], thus augmenting the measurement capabilities of cold atomic quantum simulators. These advancements may serve to enhance future explorations of the physics of cold atomic systems.

## Acknowledgments

We would like to thank Yuri van Nieuwkerk for discussions at the early stages of this work and Mohammadamin Tajik for help in using imaging resolution modelling developed by Thomas Schweigler and Frederik Møller. We also thank Tiang Bi Hong for proofreading the manuscript.

**Funding information**   TM, MG, KZA and NN were supported through the start-up grant of the Nanyang Assistant Professorship of Nanyang Technological University, Singapore which was awarded to NN. Additionally, MG has been supported by the Presidential Postdoctoral Fellowship of the Nanyang Technological University. The experiments in Vienna are supported by the Austrian Science Fund (FWF) [Grant No. I6276, QuFT-Lab] and the ERC-AdG: *Emergence in Quantum Physics* (EmQ).

## A   Free expansion dynamics

In this Appendix, we will derive the expansion dynamics of the Bosonic fields including both transversal and longitudinal dynamics, elucidating earlier works by Yuri, Essler, and Schmiedmayer [24]. Let us consider the 3D time-dependent Gross-Pitaevskii equation with zero potential

$$i\hbar \frac{\partial \Psi}{\partial t} = -\frac{\hbar^2}{2m}\nabla^2\Psi + g|\Psi|^2\Psi. \tag{A.1}$$

Upon free expansion we neglect final state interaction $g = 0$, so that the equation of motion is essentially that of free particles. The time evolution is thus given by convolution with a

Green's function

$$\Psi(\vec{r}, z, t) = \int d^2\vec{r}' \, dz \, G(\vec{r} - \vec{r}', t) G(z - z', t) \Psi(\vec{r}', z, 0), \tag{A.2}$$

where we have separated the transversal $\vec{r} = (x, y)$ and longitudinal $z$ components of the evolution and that $G(\xi, t) = \sqrt{m/2\pi i\hbar t} \exp(-m\xi^2/2i\hbar t)$ is the free, single-particle Green's function. Next, we substitute the initial state [Eq. (1) in the main text] and integrate over the transverse directions, giving us the time-evolved fields

$$\Psi_{1,2}(x, y, z, t) = \frac{1}{\sqrt{\pi\sigma_0^2(1 + i\omega_\perp t)^2}} \exp\left(-\frac{(x \pm d/2)^2 + y^2}{2\sigma_0^2(1 + i\omega_\perp t)}\right) \exp\left(\frac{im[(x \pm d/2)^2 + y^2]}{2\hbar t}\right)$$

$$\times \int dz' \, G(z - z', t) \sqrt{n_0(z')} e^{i\phi_{1,2}(z')}, \tag{A.3}$$

assuming $\omega_\perp t \gg 1$ and explicitly ignoring density fluctuation $\delta n_{1,2} \ll n_0$.

We are concerned with the density after interference of the two fields as they overlap, integrated along the vertical direction (y-axis), i.e.

$$\rho_{\text{TOF}}(x, z, t) = \int dy \, |\Psi_1(\vec{r}, z, t) + \Psi_2(\vec{r}, z, t)|^2. \tag{A.4}$$

By substituting Eq. (A.3) to Eq. (A.4), we will obtain a transverse Gaussian envelope of width $\sigma_t = \sigma_0\sqrt{1 + \omega_\perp^2 t^2}$. If we wait long enough such that $d \ll \sigma_t$ the transverse Gaussian envelopes can be approximated into a single Gaussian centred at the origin. Consequently, the expression for $\rho_{\text{TOF}}$ becomes relatively simple

$$\rho_{\text{TOF}}(x, z, t) = A e^{-\frac{x^2}{\sigma_t^2}} \left| \int_{-L/2}^{L/2} dz' \, G(z - z', t) \sqrt{n_0(z')} e^{i\phi_+(z')/2} \cos\left(\frac{kx + \phi_-(z')}{2}\right) \right|^2, \tag{A.5}$$

with $A$ being a normalization constant, $k = md/(\hbar t)$ is the inverse fringe spacing, $\phi_{\mp}(z) = \phi_2(z) \mp \phi_1(z)$ are relative (-) and common (+) phases.

## B Derivation of the transverse fit formula

We continue to derive the transversal fit formula [Eq. (4) in the main text] including the effects of mean density imbalance as well as density fluctuations. This section is a restatement of other similar derivations in the literature [22–24].

We start from the extended version of Eq. (4) in the main text, taking into account density fluctuations and different mean densities in each well

$$\rho_{\text{TOF}}(x, z, t) = A e^{-x^2/\sigma_t^2} \left| \int_{-L/2}^{L/2} dz' G(z' - z, t) e^{i\phi_+(z')/2} \left[ \sqrt{n_1(z') + \delta n_1(z')} e^{-i\phi_-(z')/2} e^{-ikx/2} \right. \right.$$

$$\left. \left. + \sqrt{n_2(z') + \delta n_2(z')} e^{i\phi_-(z')/2} e^{ikx/2} \right] \right|^2. \tag{B.1}$$

Next, we ignore longitudinal dynamics by substituting $G(z - z', t) \to \delta(z - z')$ and integrate over $z'$

$$\rho_{\text{TOF}}^\perp(x, z, t) = A e^{-x^2/\sigma_t^2} \left| \sqrt{n_1(z) + \delta n_1(z)} e^{-i\frac{kx + \phi_-(z)}{2}} + \sqrt{n_2(z) + \delta n_2(z)} e^{i\frac{kx + \phi_-(z)}{2}} \right|^2$$

$$\cong A e^{-x^2/\sigma_t^2} [n_+(z) + \delta n_+(z)][1 + C(z)\cos(kx + \phi_-(z))], \tag{B.2}$$

where

$$n_+(z) = n_1(z) + n_2(z), \qquad \delta n_+(z) = \delta n_1(z) + \delta n_2(z), \tag{B.3}$$

and interference contrast $C(z)$

$$C(z) = \frac{2\sqrt{(n_1(z) + \delta n_1(z))(n_2(z) + \delta n_2(z))}}{n_+(z) + \delta n_+(z)}. \tag{B.4}$$

Note that contrast is maximum $C(z) = 1$ when $n_1(z) = n_2(z)$ and $\delta n_1(z) = \delta n_2(z) = 0$. After absorbing $n_+(z), \delta n_+(z)$ into the normalization constant $A$, we recover Eq. (4) in the main text.

## C Corrections due to longitudinal dynamics

Here, we present a detailed derivation of the new analytical results contained in the main text [Eqs. (7) - (9)]. We start from the full expansion formula [Eq. (5) in the main text]

$$\rho_{\text{TOF}}(x, z, t) = A(x, t) \left| \int_{-L/2}^{L/2} dz' \, G(z - z', t) I(x, z', t) \right|^2, \tag{C.1}$$

where $A(x, t) = A(t) e^{-x^2/\sigma_t^2}$ and

$$I(x, z', t) = \sqrt{n_0(z')} e^{i\phi_+(z')/2} \cos\left(\frac{kx + \phi_-(z')}{2}\right). \tag{C.2}$$

We treat longitudinal expansion perturbatively by performing Taylor expansion of $I(x, z', t)$ around small $\Delta z = z' - z$

$$I(x, z', t) = I(x, z, t) + \Delta z \, \partial_z I + \frac{\Delta z^2}{2} \partial_z^2 I + \frac{\Delta z^3}{3!} \partial_z^3 I + \frac{\Delta z^4}{4!} \partial_z^4 I + O((\Delta z)^5). \tag{C.3}$$

Substituting Eq. (C.3) to the integral in Eq. (C.1), we find that the zeroth order term will give us the transversal expansion formula with a maximum contrast $C = 1$

$$\rho_{\text{TOF}}^\perp(x, z, t) \approx A(t) e^{-x^2/\sigma_t^2} \left| I(x, z, t) \int_{-\infty}^{\infty} G(\Delta z, t) \, d(\Delta z) \right|^2$$

$$= \frac{A(t) n_0(z)}{2} e^{-x^2/\sigma_t^2} [1 + \cos(kx + \phi_-(z))]. \tag{C.4}$$

Note that we have ignored boundary effects by extending the integration limit from $[-L/2, L/2]$ to $(-\infty, \infty)$. Let us compute the higher-order corrections. The first and third order terms will vanish (except near boundaries) due to the parity of the integrals. The next non-zero corrections will come from the second and fourth order terms,

$$\rho_{\text{TOF}}(x, z, t) \approx A(x, t) \left| I + \frac{\partial_z^2 I}{2} \int_{-\infty}^{\infty} (\Delta z^2 G(\Delta z, t)) \, d(\Delta z) \right|^2 + \frac{\partial_z^4 I}{4!} \int_{-\infty}^{\infty} (\Delta z^4 G(\Delta z, t)) \, d(\Delta z) \bigg|^2$$

$$= A(x, t) \left| I + i\partial_\eta^2 I - \frac{1}{2} \partial_\eta^4 I \right|^2, \tag{C.5}$$

with $\eta = z/\ell_t$ is dimensionless coordinate and $\ell_t = \sqrt{\hbar t/(2m)}$. Next, we approximate Eq. (C.5) by including terms up to the fourth order in derivatives of $I$

$$\rho_{\text{TOF}}(x, z, t) \approx \rho_{\text{TOF}}^\perp(x, z, t) + \Delta\rho^{(2)} + \Delta\rho^{(4)} \tag{C.6}$$

$$= A(x, t) \left[ |I|^2 - 2 \, \text{Im}(I^* \partial_\eta^2 I) + |\partial_\eta^2 I|^2 - \text{Re}(I^* \partial_\eta^4 I) \right], \tag{C.7}$$

with $\Delta\rho^{(n)}$ being the n-th order correction terms in scaled derivatives $\partial_\eta I$. We first focus on the leading order correction $\Delta\rho^{(2)} = -2A(x,t)\mathrm{Im}(I^*\partial_\eta^2 I)$. To obtain this, we first compute $\partial_z^2 I$

$$\partial_z^2 I = \Gamma(z)\cos\left(\frac{kx + \phi_-(z)}{2}\right) - \Lambda(z)\sin\left(\frac{kx + \phi_-(z)}{2}\right), \tag{C.8}$$

where

$$\Gamma(z) = \partial_z^2\psi_+ - \frac{\psi_+(\partial_z\phi_-)^2}{4}, \qquad \Lambda(z) = \partial_z\psi_+\partial_z\phi_- + \frac{\psi_+\partial_z^2\phi_-}{2}, \tag{C.9}$$

and $\psi_+(z) = \sqrt{n_0(z)}e^{i\phi_+(z)/2}$. For simplicity, we will consider the case $n_0(z) = n_0 = \mathrm{const.}$ which gives us

$$\Delta\rho^{(2)} = -A(x,t)\frac{n_0}{2}[\partial_\eta^2\phi_+(1 + \cos(kx + \phi_-)) - \partial_\eta\phi_-\partial_\eta\phi_+\sin(kx + \phi_-)]. \tag{C.10}$$

Combining the above with the expression for $\rho_{\mathrm{TOF}}^\perp$ in Eq. (C.4) and using trigonometric identity $a\cos x + b\sin x = \sqrt{a^2 + b^2}\cos(x - \alpha)$ with $\tan\alpha = b/a$ we can express $\rho_{\mathrm{TOF}}$ as

$$\rho_{\mathrm{TOF}}(x,z,t) \approx A'(z,t)e^{-x^2/\sigma_t^2}[1 + C(z,t)\cos(kx + \phi_-(z) - \Delta\phi_-^{(2)}(z,t))], \tag{C.11}$$

with

$$A'(z,t) = \frac{A(t)n_0}{2}\left(1 - \partial_\eta^2\phi_+\right), \tag{C.12}$$

$$C(z,t) = \frac{1}{\left(1 - \partial_\eta^2\phi_+\right)}\sqrt{\left(1 - \partial_\eta^2\phi_+\right)^2 + \left(\partial_\eta\phi_-\partial_\eta\phi_+\right)^2}, \tag{C.13}$$

$$\Delta\phi_-^{(2)}(z,t) = \arctan\left(\frac{\partial_\eta\phi_+\partial_\eta\phi_-}{1 - \partial_\eta^2\phi_+}\right) \approx \partial_\eta\phi_+\partial_\eta\phi_-. \tag{C.14}$$

Eq. (C.14) is the first term of Eq. (9) in the main text. We are also interested in cases where $\phi_+ = 0$. For such cases, the second-order term vanishes and so we turn to the fourth-order terms

$$\Delta\rho^{(4)} = A(x,t)\left[|\partial_\eta^2 I|^2 - \mathrm{Re}(I^*\partial_\eta^4 I)\right]. \tag{C.15}$$

We will only calculate this term for uniform gases with zero common phase case so that $I(x,z,t) = \sqrt{n_0}\cos((kx + \phi_-(z))/2)$ is real. After a straightforward but lengthy calculation, invoking essentially the same trigonometric trick as before, we obtain subdominant phase shift

$$\Delta\phi_-^{(4)}(z,t) \approx -\frac{1}{2}(\partial_\eta\phi_-)^2(\partial_\eta^2\phi_-). \tag{C.16}$$

This is the second term of Eq. (9) in the main text.

# D Relative phase fitting initialization

In this section, we show the approximate linear relationship between relative phase $\phi_-$ and the interference peak's transversal position $x_{\max}$ for a fixed longitudinal position $z$. We use this approximate linear relationship to provide an initial guess for the optimizer used in fitting.

For simplicity, we assume $\rho_{\mathrm{TOF}}$ to be well approximated by the standard fitting formula [Eq. (4) in the main text] with $C = 1$. To find the transversal peak location, we simply solve $\partial\rho_{\mathrm{TOF}}^\perp/\partial x|_{x=x_{\max}} = 0$, which gives the condition

$$\frac{2x}{\sigma_t^2}\left[1 + \cos\left(kx_{\max} + \phi_-^{(0)}\right)\right] + k\sin\left(kx_{\max} + \phi_-^{(0)}\right) = 0, \tag{D.1}$$

where the superscript 0 indicates a 'guess' value (initial value to feed into the optimizer). Using the half-angle formula, we obtain

$$\cos\left(\frac{kx_{\max}+\phi_-^{(0)}}{2}\right)\left[\frac{2x}{\sigma_t^2}\cos\left(\frac{kx_{\max}+\phi_-^{(0)}}{2}\right)+k\sin\left(\frac{kx_{\max}+\phi_-^{(0)}}{2}\right)\right]=0\,. \qquad \text{(D.2)}$$

For non-zero interference, we must have $\cos([kx+\phi_-^{(0)}]/2)\neq 0$ and so to satisfy Eq. (D.2), the terms inside the parenthesis have to vanish. Finally, we can solve for $\phi_-^{(0)}$ and the result is

$$\phi_-^{(g)}=-kx_{\max}+2\arctan\left(-\frac{2\omega_\perp t}{1+\omega_\perp^2 t^2}\frac{x_{\max}}{d}\right)\approx-\frac{md}{\hbar t}x_{\max}\,, \qquad \text{(D.3)}$$

where in the last approximation we have used $\omega_\perp t\gg 1$ such that the arctan function changes very slowly with $x_{\max}$.

# E   Supplementary plots

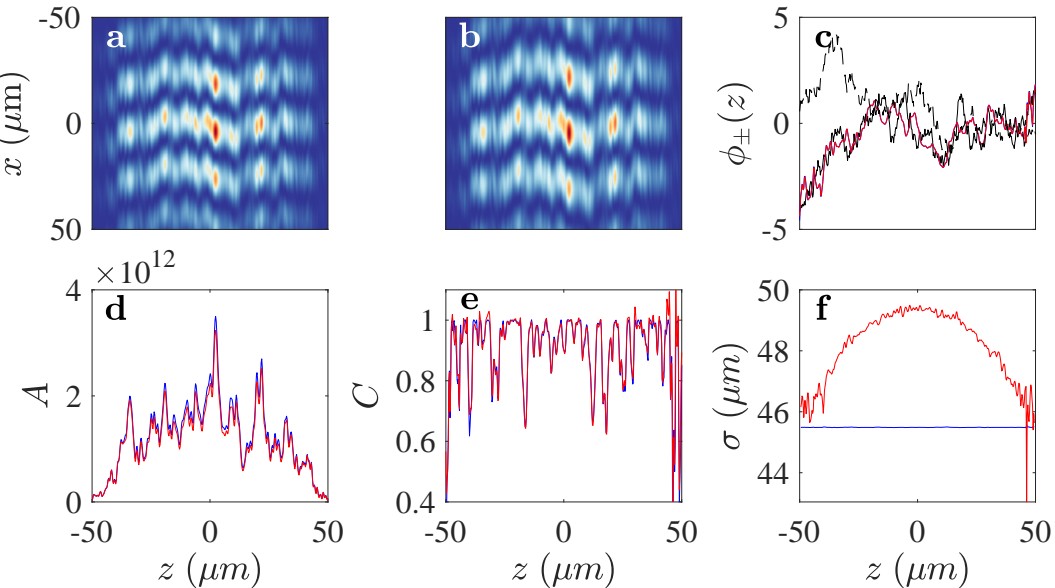

Figure 15: Single shot relative phase extraction with and without broadening due to atomic repulsion. Panels **a** - **b** show interference pattern without ($\sigma_0=$ const.) and with scattering-induced broadening ($\sigma_0^2(z)=\sigma_0^2\sqrt{1+a_s n_0(z)}$). Panels **c-f** show the extracted fit parameters $\{\phi_-,A,C,\sigma\}$ without (red) and with interaction broadening. The black solid (dashed) line in panel **c** is the input relative (common) phase. From this figure, we observe that scattering-induced broadening does not significantly impact the extracted fit parameters except the width.

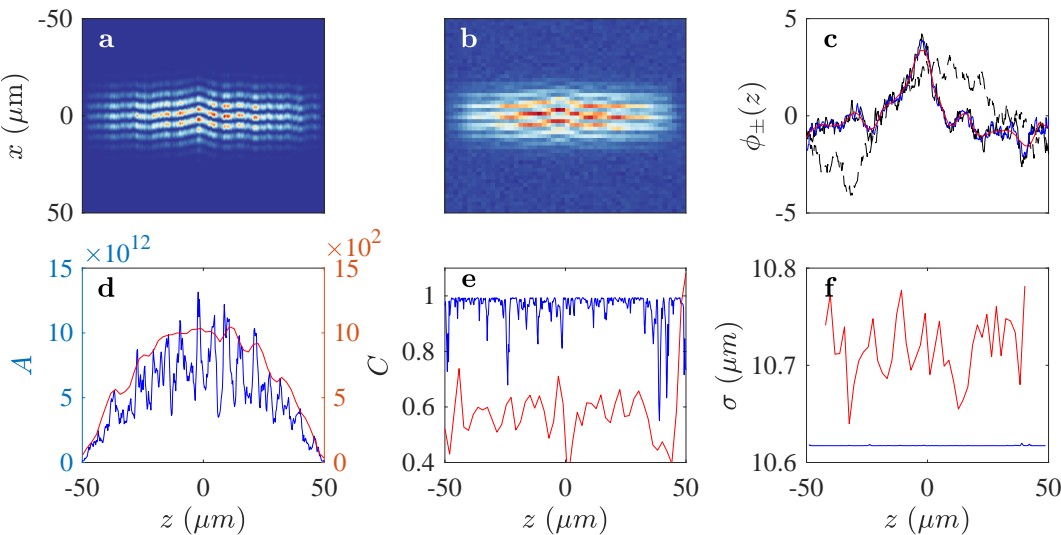

Figure 16: Single shot relative phase extraction with (red) and without (blue) imaging effects, similar to that of Figure 13 but for a short time-of-flight $t = 3.5$ ms. The common phase is denoted by a dashed line in panel **c**.

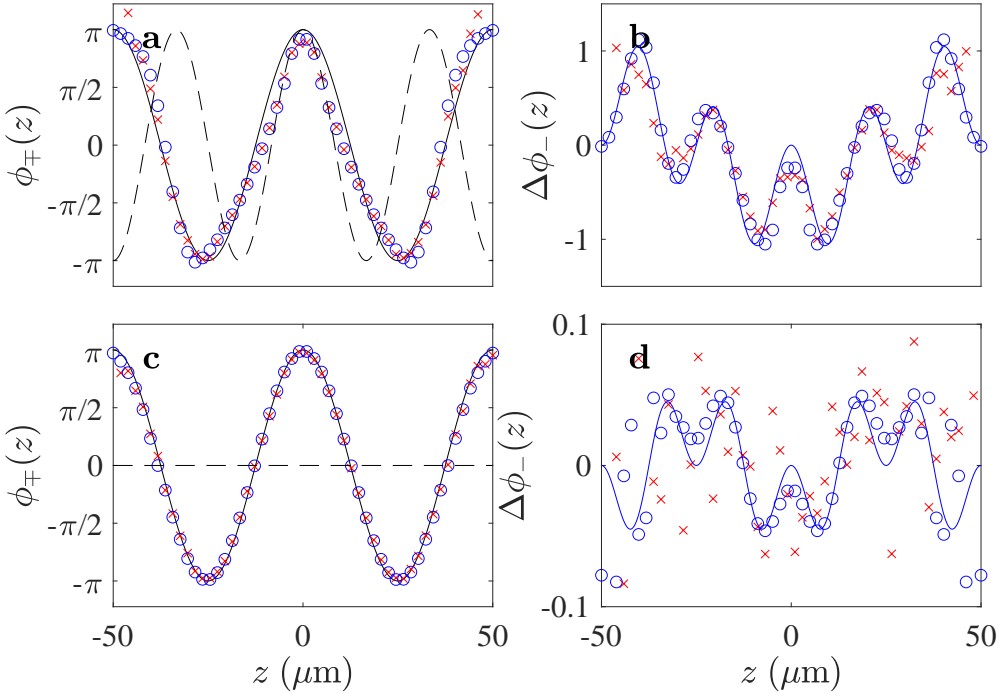

Figure 17: Simulation of analytic systematic phase shift with (red) and without (blue) imaging effects for 15 ms expansion time, see Fig. 3 for comparison.

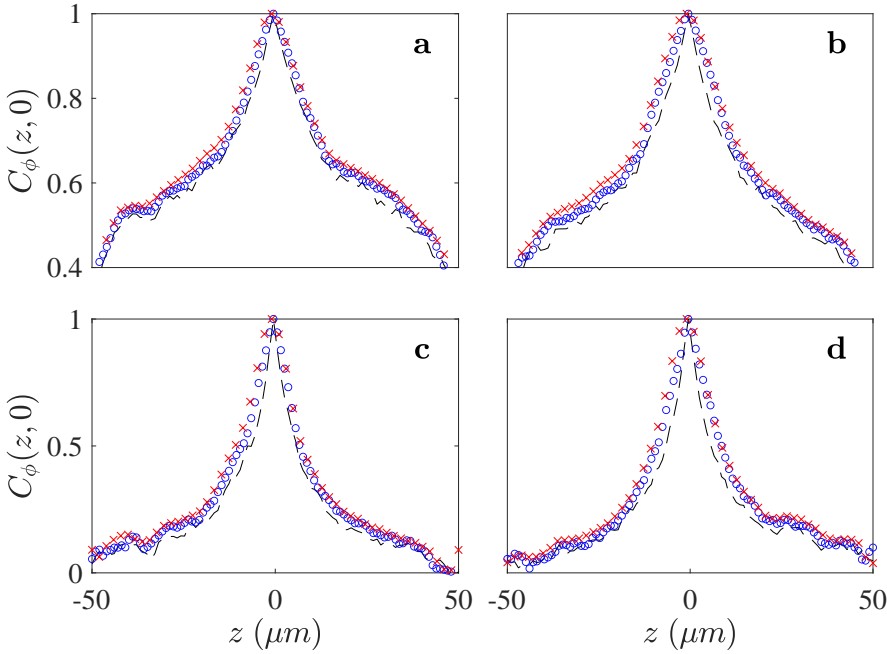

Figure 18: Vertex correlation function $C_\phi$ with (red crosses) and without (blue circles) image processing. Each panel corresponds to the same parameter regime and number of shots as in Fig. 5 except that the common phase is always sampled from a thermal state with $T_+ = T_-$. The statistics are obtained with the camera defocusing set to 0 (ignoring recoil and free falling of the cloud during exposure), but we expect defocusing effect to be small.

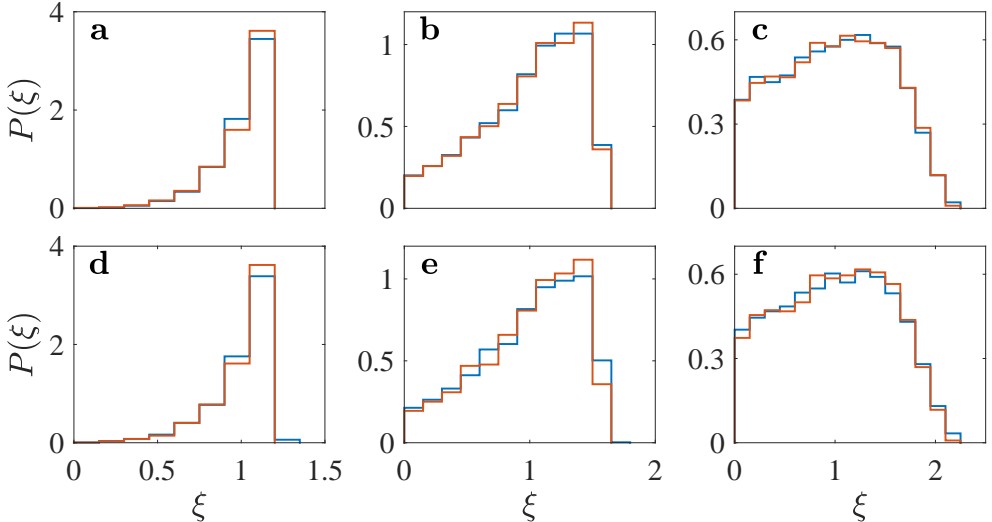

Figure 19: Full distribution function $P(\xi)$ with (red) and without (blue) imaging effects reconstructed with 5000 TOF simulations. Each panel correspond to the same parameter regime as Fig. 6 except for a fixed expansion time $t = 15$ ms. The statistics are obtained with the camera defocusing set to 0 (ignoring recoil and free falling of the cloud during exposure), but we expect defocusing effect to be small.

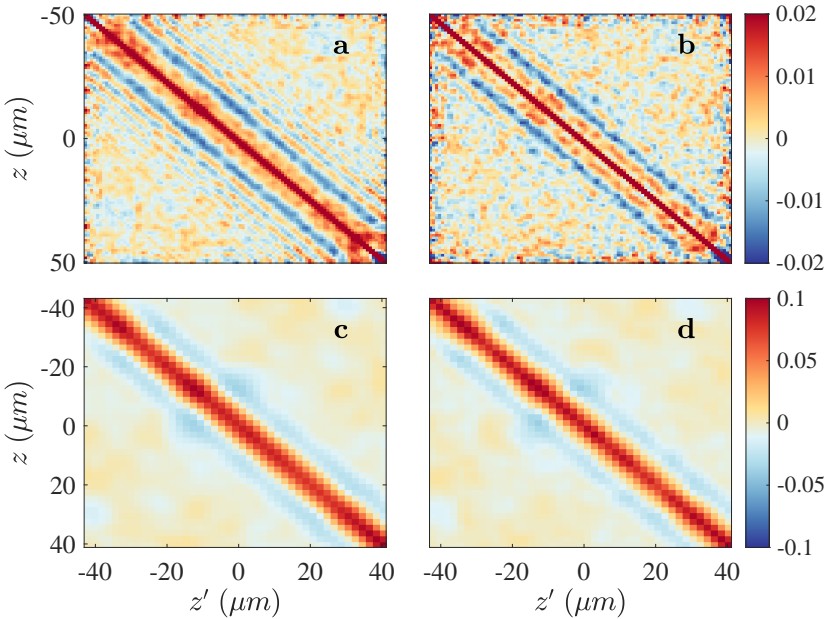

Figure 20: Velocity-velocity correlation $C_u(z, z')$ without (**a-b**) and with (**c-d**) imaging effects. The first column (**a**, **c**) corresponds to the case with $\phi_+(z) = 0$ and the second column (**b**, **d**) corresponds to the case with common phase sampled from the same thermal distribution as the relative phase $T_+ = T_- = 75$ nK. The statistics are obtained with the camera defocusing set to 0 (ignoring recoil and free falling of the cloud during exposure), but we expect the effect of defocusing to be small. The edge data of length 5 $\mu$m on each end have been omitted to suppress boundary effects.

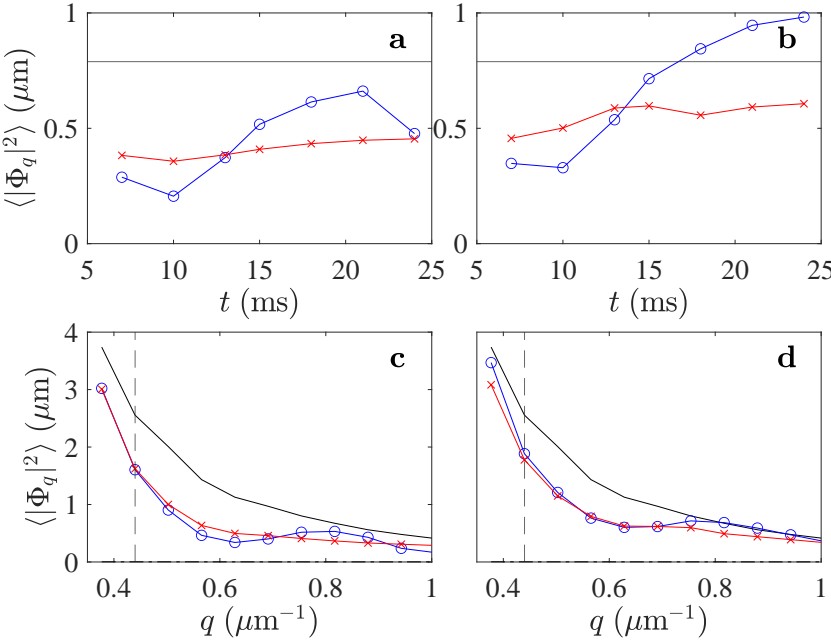

Figure 21: TOF reconstruction of mean Fourier coefficients (solid black line) without (blue circles) and with (red crosses) imaging effects computed with 500 TOF simulations for $T_- = 75$ nK. The solid black line represents the input value(s). In panels (**a**)-(**b**), the mode is fixed at $q = 22\pi/L \approx 0.69$ $\mu m^{-1}$ while the expansion time $t$ is varied. In **c**-**d** $t$ is fixed at 15 ms but $q$ is varied. The dashed vertical line at $q \approx 0.44$ $\mu m^{-1}$ indicates the point where deviation due to imaging is apparent. The horizontal dashed-dot line shows the shot-noise fluctuations computed with TOF simulations of $\phi_-(z) = 0$. The first column (**a,c**) is for the case with $\phi_+(z) = 0$, the second column (**b,d**) is for the case with common phase sampled from thermal state with $T_+ = T_-$. The statistics are obtained with camera defocusing set to 0 (ignoring recoil and free falling of the cloud during exposure), but we expect the defocusing effect to be small.

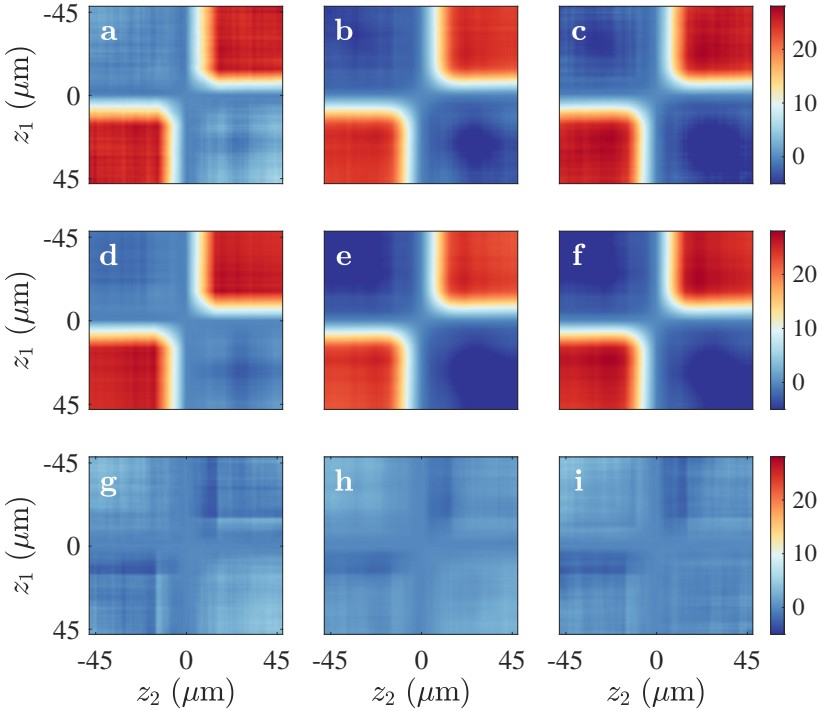

Figure 22: time-of-flight (TOF) reconstruction of fourth-order correlation function $G^{(4)}(z_1, z_2, z_3, z_4)$ of the sine-Gordon model in the Gaussian regime ($\chi = \lambda_T / l_J = 0.5$). The data is cut at $z_3 = -z_4 = 15$ $\mu$m for visualization. Panels (**a-c**) show the full correlation, panels (**d - f**) show the disconnected part, and panels (**g - i**) shows the connected part. The first column (**a, d, g**) represents the case with only transversal expansion, the second column (**b, e, h**) includes longitudinal expansion but with common phase kept at zero and the last column (**c, f, i**) corresponds to the case with longitudinal expansion and common phase sampled from thermal distribution with $T_+ = 75$ nK. Each panel is reconstructed from 2500 TOF realizations with 15 ms expansion time. The edge data of length 2.5 $\mu$m on each end have been omitted to suppress boundary effects.

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
