# Peer review of "Systematic analysis of relative phase extraction in one-dimensional Bose gases interferometry"

_SciPost Physics, doi:SciPost Phys. 18, 065 (2025)_

## Round 1 · Referee Report · Anonymous (Referee 2) · 2024-11-10

Strengths

  1. Introduces improvements of currently used method of relative phase measurement.
  2. Provides a detailed derivation and strong arguments supporting the new formula for phase estimation.
  3. Reveals differences in using the full expansion formula with respect to transversal fit formula in a number of physical quantities.
  4. Identifies approximations assumed in the new formula, whose impact will hopefully be addressed in a future work.

Weaknesses

-

Report

This work addresses one of the fundamental measurement in the field of ultracold gases, namely the relative phase extraction from interference patterns between freely expanded matter waves. The authors revised currently used methods used to estimate the phase and proposed an extended, more accurate formula, which includes previously neglected influence of longitudinal expansion on the result of measurement. The 'full expansion formula' (eq. 5) derived in this work is then compared to the 'transversal fit formula' in a reconstruction of different physical quantities.
In general I find the manuscript to be well written and organized. Results of this work are of a high importance, especially for improvement of experimental measurements. Therefore, I recon this work should be published in a high-impact journal.

Recommendation

Publish (easily meets expectations and criteria for this Journal; among top 50%)

  • validity: top
  • significance: high
  • originality: high
  • clarity: top
  • formatting: excellent
  • grammar: excellent

Author:  Nelly Ng  on 2025-01-15  [id 5122]

(in reply to Report 1 on 2024-11-10)

Reply: We thank the referee for the positive comments on our manuscripts and for the recommendation to publish.

We include here also a list of other important changes, which are not made in direct response to any points from referees, but are done to improve the manuscript: - The first two paragraphs of the introduction have been slightly rewritten for clarity - Eq. (9) has been slightly modified, with some of the constant prefactors absorbed into the definition of $\ell_t$. More importantly, the sign of the second term is changed into negative. - (rederivation to correct a sign error; other results unchanged) Theoretical derivation in Appendix C has been updated to include the fourth order Taylor expansion, which is important for getting the right constant prefactor in the second term of Eq. (9). - The title of Sec. 4.1 is changed from “Two-point phase correlation function” to “Vertex correlation functions” to distinguish it from the two-point correlation function discussed in Sec. 4.5. The relevant inline text and figure caption have been changed accordingly. - In Sec. 4.3 last paragraph, we added “To resolve such an effect, one must resolve fluctuation with a length scale comparable to the length scale of TOF dynamics $\ell_t = \sqrt{\hbar t/(2m)}$. Thus, this effect might not be captured by present experiments and by our perturbative treatment in Eq. (9). However, our numerical results point to the necessity of calibrating the results of dynamical propagation of velocity-velocity correlation such as in Ref. [19] to the measurement background in future experiments with enhanced resolution.”

---

## Round 1 · Referee Report · Anonymous (Referee 3) · 2024-12-2

Strengths

1.This paper revisits the study of extraction of the relative phase between two atomic Bose-Einstein condensates under expansion, characterizing the role of longitudinal dynamics during the process.
2.The manuscript is clearly presented, sound and rather extensive.
3.The manuscript is presented in a manner which could be useful to experimental group analysis in the future.

Weaknesses

None

Report

This is an interesting and detailed manuscript investigating the role of longitudinal dynamics during the extraction of the relative phase of two one-dimensional Bose gases. The papers is clearly set in the existing literature, and its aims are very clear. The paper performs a detailed analysis of the impact on different measurable quantities, and also comments on issues related to image processing, making it a very thorough study.

I recommend acceptance of this manuscript.

Requested changes

The only comments I have (after having also seen the other referee report and author response) are the following:

1.How important would the authors expect the role of density fluctuations (ignored in this work) be, and how far do they believe the current findings can be pushed under experimental conditions in the presence of density fluctuations? Perhaps some further discussion on this would be useful.

2.Given that expansion imaging is the standard method of extraction of information, could the authors perhaps comment (more) on the relevance of their findings to: (i) Ref. [32] (which is duplicated as Ref. [42]): in particular, and noting the existing analysis, would the authors still expect the originally published results to hold reasonably (for all probed observables, most notably the higher ones)? I think the discussion / re-interpretation of that work could have been more extensive/transparent (i.e. whether any key findings are now being cast into question). (ii) https://doi.org/10.1103/PhysRevLett.130.123401 studied the 1D-3D crossover through expansion imaging and extremely thorough numerics. Can the present analysis also say something about such work? Is there anything that might, for example require re-interpretation?

  1. Finally, I was wondering whether the authors were aware of the below manuscript, and -- if not -- whether they thought that some of the presented analysis might be useful in their work: Piotr Deuar, Comput. Phys. Commun. 208, 92 (2016) 10.1016/j.cpc.2016.08.004 (arXiv:1602.03395 ) A tractable prescription for large-scale free flight expansion of wavefunctions

Recommendation

Publish (easily meets expectations and criteria for this Journal; among top 50%)

  • validity: high
  • significance: high
  • originality: high
  • clarity: high
  • formatting: excellent
  • grammar: perfect

Author:  Nelly Ng  on 2025-01-15  [id 5121]

(in reply to Report 2 on 2024-12-02)

Reply: We thank the referee for the positive feedback on our manuscript, including the valuable suggestions and the recommendation to publish.

  1. At the time we wrote the manuscript, in-situ density fluctuation was omitted to simplify the analysis and the simulation procedure. Since then, we have performed a preliminary analysis in which the in-situ density fluctuation is included in the simulation. More precisely, we rerun some of the numerical simulations in this paper, taking into account in-situ density fluctuations. For example, the key results in Fig. 3 are virtually unchanged by the presence of in-situ density fluctuation. Therefore, we expect in-situ density fluctuations to play only a minor role in the systematic error due to longitudinal expansion, which is also the main message of this paper.

  2. (i) We first thank the referee for pointing out the duplicated reference. We have removed the duplicated copy of the reference in the updated manuscript. In the first paragraph of Sec. 6, we stated: “We find that Gaussian observables and two-point correlations are well-preserved by the TOF measurement since those depend mostly on low momentum phase fluctuation. However, for higher moments and observables sensitive to high momentum fluctuation, we observe that TOF introduces some deviations so they must be taken with great care.” In the updated manuscript, we elaborate on this point for more clarity on our result: “We find that Gaussian observables (vertex correlation function, two-point correlation function, full distribution function) are well-preserved by time of flight. However, for higher moments and for observables sensitive to high-momentum fluctuations, e.g. mean Fourier spectrum, velocity-velocity correlation, and fourth-order correlation functions, deviations arise due to longitudinal dynamics and so they must be taken with great care. In the case of mean Fourier coefficients and velocity-velocity correlation, the deviations are mostly due to dynamics in the high momentum modes, which lie beyond our perturbative analytical treatment and current experimental imaging resolution. However, these deviations might still be important to consider in future experiments with improved resolution. For the fourth-order correlation functions, local deviation persists even after taking into account current experimental imaging resolution. Whether our perturbative correction formula can be used to correct this deviation and to what extent this deviation affects the measure of non-Gaussianity [33] will be explored in future work.” (ii) We thank the referee for pointing us out to the Ref. https://doi.org/10.1103/PhysRevLett.130.123401. In our understanding, the author of the above paper probes dimensionality cross-over by tuning chemical potential $(\mu)$ and temperature $(T)$ as compared to the energy scale of the trap $(\hbar \omega_\perp)$. In our case, we restrict ourselves to 1D, where $\mu, k_B T << \hbar \omega_\perp$. The paper also focuses on a single condensate system, whereas our paper focuses on interference between two 1D systems. The paper probes the system by measuring the spectrum of density ripple after time of flight, which for 1D systems occurs due to longitudinal dynamics. In our case, longitudinal dynamics also induce density ripple in the interference pattern as demonstrated in Fig. 2. In the present work, we do not consider density ripple and instead focus on the correction to the relative phase readout. However, in our recent preprint (https://arxiv.org/html/2408.03736v2), we demonstrate how to infer the phase (or common phase) of such 1D systems from density ripple information.

  3. We thank the referee for pointing us out toward this paper. We believe that this paper is relevant to our work and therefore we have cited it in the updated manuscript (Ref. [29]). The main concern of the Comput. Phys. Commun. 208, 92 (2016) is on how to efficiently choose the lattice grid for simulating TOF expansion. Given that during the TOF, the gas can expand up to 10 to 100 times its initial size, the full 3D simulation of this process can use a lot of memory. In our case, we circumvent this problem by making use of the fact that we have a 1D system, and therefore, assuming fully ballistic expansion, the solution for transverse dynamics is known exactly. However, the above paper will be relevant for the full simulation of TOF expansion which includes final state interaction. In our present work, this stage is ignored. We believe this is a reasonable assumption given that the pre-ballistic regime occurs in a timescale of $1/\omega_\perp \sim 0.1$ ms, whereas in experiments we are interested in the timescale of $\sim 10$ ms.

---

## Round 1 · Referee Report · Marc Cheneau (Referee 4) · 2024-12-11

Report

This report was written jointly by Marc Cheneau and Isabelle Bouchoule (Laboratoire Charles Fabry).

In this paper, the authors investigate the extraction of the relative phase between two 1D Bose gases from the analysis of the interference pattern which appears after a time of free expansion, when the two matter waves overlap. This technique has been used in many previous works of the Vienna group led by J. Schmiedmayer, coauthor of of the present article. In these works, the longitudinal motion during the time of flight was neglected, and the transverse position of the fringes at a given longitudinal position $z$ was identified with the initial relative phase between the two gases at the same position $z$.

Here, the authors instead focus on this longitudinal motion. The authors first explain how they can analytically account for the longitudinal motion, and they clearly explain the underlying assumptions. Then, making an expansion in the length associated with the time-of-flight, they derive a formula showing that the first dominant effect of the longitudinal motion is a coupling between the relative and common phases, while the second sub-dominant effect involves the first and second order derivatives of the relative phase along $z$. The rest of the paper is a numerical investigation based on the aforementioned formula. More precisely, the authors quantify the error made if the interference pattern is analysed ignoring the longitudinal motion during the time-of-flight---as was done previously. For this analysis, the authors consider two clouds initially at thermal equilibrium, both with and without Josephson coupling. Different situations are considered, which correspond to different experimental studies published by the Vienna group in the past. In most cases, the effect of longitudinal motion during the time-of-flight is found to be negligible, unless very small length scale are considered---which are usually out of reach anyway because of the finite resolution of the imaging system.

Our assessment is that this article is sound, pedagogical (for the sections 2 and 3), and addresses an effect which had not been previously studied. Given that the analysis of matterwave interference patterns is a widely spread tool to access subtle physical phenomena, investigating in detail the limitations and the domain of validity of the method employed previously is of high importance. We therefore think that this article deserves publication in SciPost.

Still, we have several comments which we would like to share with the authors. We would also like to draw the authors attention on the presence of some grammatical mistakes, among which the frequent absence of a definite article (“the”). Also, the discussion in section 4 is sometimes hard to follow.

  1. On p. 2, the statement “If the trap is switched off rapidly, the dynamics are well approximated by a quench into free evolution” should probably include the requirement of a rapidly decreasing density, which allows one neglecting the effect of interactions during the expansion.

  2. Before Eq. (6), the authors say that Eq. (5) can hardly be used as a fit function. However, intuitively, on can a priori extract the phases $\phi_+$ and $\psi_-$ from the interference pattern. The reason is that the interference pattern contains 3 z-depend functions---a linear total density, a contrast $C'$ and a relative phase $\phi_{\rm mes}(z)$---while, assuming that the initial density fluctuations are negligible, there are only two z-dependent functions to retrieve. This issue is discussed a bit in the conclusion but the authors might want to elaborate on this a bit more.

  3. From Eq.(5), it is clear that for a given $z$, one expects a density pattern of the form given by Eq.(7), although $\Delta \phi_-$ might not be given by Eq. (9). This form is not restricted to the second order expansion made to derive Eq.(9). We therefore don't think that it is appropriate to write, after Eq. (7): “This demonstrates the robustness of the transverse fit formula.” This transverse fit formula is exact, within the approximations made to derive Eq. (5), and it does not depend on the expansion of $I$.

  4. The formula in Eq. (9) is very nice and it is a that pity the authors do not propose to use it to give analytical predictions for clouds at thermal equilibrium. Then one could fit the interference pattern with 2 temperatures: one for the common modes and one for the relative modes. It seems that predictions for the situations considered in this paper could be done using this formula. Such a prediction could be compared to numerical calculation to test the second order expansion approximation.

4bis. The caption of Fig. 3 says that the solid lines are fitting curves based on Eq. (9). Why fitting the data instead of computing the prediction from Eq. (9) using the known input parameters?

4ter. As explained in the text, Eq.(9) is only valid if one considers wavelengths much larger than $\hbar/\sqrt{t}$. It does not capture phenomena involving wavelengths of the order of, or smaller than $\hbar/\sqrt{t}$. In particular it does not capture the oscillating feature investigated in section 4.4.

4four. Eq. (9) is tested numerically in Fig. 3. However, it is not used in the rest of the paper, when the reconstruction of physical quantities are considered. We think it would be instructive to compar...

  1. In the caption of Fig. 3, we are not sure to understand the meaning of the sentence “Numerical errors have been accounted for by subtracting the phase error using the transversal model in both encoding and decoding.” Could the authors maybe rephrase their remark?

  2. Below the title of section 4, the initial state of the system is said to be given by a certain Hamiltonian. The authors probably means that it is given by the ground state of this Hamiltonian.

  3. In Eq. (11), $\delta \rho$ is not defined.

  4. On p. 8, we found the description of the method used to independently sample the relative and common phase profiles difficult to understand without looking into the given references.

  5. In section 4.2, the authors discuss the retrieval of the the full distribution function $P(\xi)$, as done in ref. [13]. Following this reference, they claim that $P(\xi)$ “provides unambiguous signatures of quantum fluctuations”. However we know from [Mazets, Phys. Rev. Lett. 105, 015301 (2010)] that the measurements reported in [13] are actually compatible with thermal expectations.

  6. In section 4.4, the authors use the terminology “mean occupation number” for the Fourier spectrum of the phase. We think that this terminology is misleading since it conveys the idea of an occupation number for a quantum bosonic mode, which is not the case here: the model is fully classical without any zero-point of motion and quantization of its energy. This approach is relevant only for experiments where the modes which are probed are highly occupied.

  7. In section 4.4, the notation $k$ for the longitudinal Fourier space coordinate of the phase might be confusing because this symbol is already used to denote the wavevector of the interference pattern in the vertical direction. The authors should probably use another symbol, like $q$.

  8. The right-hand side of Eq. (15) has the dimension of the a length, whereas, according to the definition given above Eq. (15), $\langle|\phi_k|^2\rangle$ is dimensionless. In this definition, the factor $1/L$ should probably be replaced by $1/\sqrt{L}$.

  9. If the phase fluctuations present in the initial system are Gaussian, then the same information is contained in $\langle\cos(\psi_-(z)-\psi_-(z'))\rangle$ (section 4.1) and in $\langle|\phi_k|^2\rangle$ (section 4.4). The main difference between the two observables is that the first one mixes different modes, while the second one treats each mode separately. The authors might want to mention this subtlety in their article, and comment on the relative merit of both observables..

  10. As far as we understand, the oscillations of a given Fourier component discussed in section 4.4 are not captured by the correction to the phase given in Eq.(9). Indeed, let us assume that initially only a single mode of wavevector $q$ is present, with a low amplitude, such that the semiclassical field is approximately given by the sum of a constant and a term proportional to the phase:

    $$\psi_{1/2}(z )\simeq \sqrt{n} \pm i\sqrt{n}\Theta\cos(qz),$$
    with $\Theta \ll 1$. The relative phase pattern before the time-of-flight is thus
    $$\theta_-(z,t=0)=\Theta \cos(qz).$$
    Then, one finds that the relative phase between the two clouds after the time-of-fight is
    $$\theta_-(z,t)=\Theta \cos(qz)\cos(\hbar^2q^2/(2t)).$$
    Thus one finds that $|\phi_q|^2$ oscillates as a function of the time-of-flight, similar to the Talbot effect in optics. Such an oscillation is not captured by Eq.(9), which is of higher order in the phase, and which results from the non linearity of the atomic field with respect to the phase. Eq.(9) is only valid if one consider wavelengths much larger than $\hbar/\sqrt{t}$. We think that this could be explained in the text.

  11. In section 4.5, the authors investigate the extraction of higher order correlation functions of the relative phase for a system described by the Sine-Gordon Hamiltonian. The authors speak about “many-body problem”. We don't think that this terminology is appropriate here since, in the context of this work, the physics is that of a classical field whose Hamiltonian---the Sine-Gordon Hamiltonian---is non linear in the field $\phi_-$. We would therefore rather say “non linear field theory”.

  12. In section 4.5, when analyzing the 4th-order correlation function, the authors explain that “the disconnected part appears to be considerably modified by TOF”. We don't understand how this disconnected part, which involves only a second-order correlator, can be “considerably modified” when the changes in the second-order correlation itself are modest.

  13. The value of the parameter $q$ should be indicated in the legend of Figs. 10 and 11.

  14. We find the term “image processing” (title of section 5 plus a few other places) misleading. This term usually refers to the analysis or treatment of a digital image, while the authors rather think of how the the imaging process affects the measured density distribution.

Recommendation

Ask for minor revision

  • validity: high
  • significance: good
  • originality: good
  • clarity: ok
  • formatting: good
  • grammar: below threshold

Author:  Nelly Ng  on 2025-01-15  [id 5120]

(in reply to Report 4 by Marc Cheneau on 2024-12-11)
Category:
answer to question
correction
validation or rederivation

We thank the referee for the positive feedback on our manuscript, including many detailed and valuable suggestions and the recommendation to publish.

  1. As suggested by referee, we have changed the sentence on p. 2 “If the trap is switched off rapidly, the dynamics are well approximated by a quench into free evolution” into “If the trap is switched off rapidly, the initial tight confinement in transverse directions leads to rapidly expanding density, which allows one to neglect the effect of interactions during the expansion. Consequently, the dynamics are well approximated by a quench into free evolution.”

  2. “Intuitively, on can a priori extract the phases ϕ+ and ψ− from the interference pattern. The reason is that the interference pattern contains 3 z-depend functions---a linear total density, a contrast C′ and a relative phase ϕmes(z)---while, assuming that the initial density fluctuations are negligible, there are only two z-dependent functions to retrieve.” At the time of writing of this manuscript, to the best of our knowledge, there was no reliable way to extract $\phi_+$ information from interference patterns. However, our recent preprint https://arxiv.org/abs/2408.03736 shows how to do exactly this from density ripple. As pointed out by the referee, this points to the possibility of using the full formula (Eq. (5)) to fit the density interference pattern (ignoring in situ density fluctuation). But, we believe this is still not simple due to the presence of the integral, which means that one has to guess the entire function $\phi_-(z)$ and calculate the integral $\rho_{TOF}$ at every iteration. This can be numerically costly and it is not even clear whether such an optimization problem is convex. This type of ‘functional’ optimization as a fitting process is beyond the scope of this work. An alternative is to invert Eq. (5) in Fourier space instead of in real space. These possibilities will be explored in future work.

  3. As suggested by referee, we have removed the following sentence below Eq. (7) “The above implies that, at least up to the second order, longitudinal expansion does not change the functional relationship between $\rho_{\rm TOF}(x,z,t)$ and $\phi_-(z)$. This demonstrates the robustness of the transversal fit formula; nevertheless, longitudinal expansion still influences the extracted fit parameters. “ and replace it with “The form of Eq. (7) is expected from Eq. (3). Using our integrand expansion technique, we are able to express the fit parameters in terms of in-situ field variables.”

  4. “The formula in Eq. (9) is very nice and it is a that pity the authors do not propose to use it to give analytical predictions for clouds at thermal equilibrium. Then one could fit the interference pattern with 2 temperatures: one for the common modes and one for the relative modes. It seems that predictions for the situations considered in this paper could be done using this formula. Such a prediction could be compared to numerical calculation to test the second order expansion approximation. ” Eq. (9) relates in situ relative phase and readout relative phase from interference pattern. In experiment, the only variable we have access to is the readout relative phase. So, we are not sure what the referee means by using the formula (9) to fit temperatures. Temperatures of the common phase can be extracted not with Eq. (9), but by looking into density ripple (see also our other preprint https://arxiv.org/abs/2408.03736). We are interested in applying Eq. (9) to perform single-shot corrections to relative phase extraction in thermal equilibrium, especially now that we understand how to extract the common phase. But, the challenge we see in applying Eq. (9) to thermal fluctuations is that when calculating the derivatives, we might include noise coming from high momentum fluctuations beyond TOF cutoff $q ~ \ell_t^{-1}$. This will also be explored in future work. “The caption of Fig. 3 says that the solid lines are fitting curves based on Eq. (9). Why fitting the data instead of computing the prediction from Eq. (9) using the known input parameters?” We thank the referee for this suggestion. Previously, we fitted the constant prefactors in front of the derivatives. However, our Eq. (9) indeed allows us to calculate the prefactor. In the process of doing this, we discover a small error: the second term of Eq. (9) should have a minus sign. We have corrected this. We also modify the prefactor slightly by absorbing some of the constants to the definition of $\ell_t$. In the updated submission, we have changed Fig. 3 and its caption to show that not only we are able to predict the functional form of $\Delta \phi_-$, we also obtain the correct constant prefactors.

  5. “In the caption of Fig. 3, we are not sure to understand the meaning of the sentence “Numerical errors have been accounted for by subtracting the phase error using the transversal model in both encoding and decoding.” Could the authors maybe rephrase their remark?” We have removed this sentence in the updated submission. Previously, we anticipated some error to arise from the optimizer, which has nothing to do with TOF systematics. We control this by performing encoding and decoding of relative phase purely with the model that considers only transversal expansion. However, we checked that such numerical error is extremely small << 0.1 rad. We therefore ignore this in the new figure and omit the sentence from the caption.

  6. We have added “thermal state of the sine-Gordon Hamiltonian….” below the title of section 4.

  7. We have changed $\delta \rho$ to $\delta n$ to be consistent with previously defined notation for in situ density fluctuations.

  8. In sampling the input fluctuations, we perform sampling with multivariate normal random distribution from a theoretically computed covariance matrix. This is just one way of obtaining input fluctuations, another equivalent approach is described in Stimming et al. Phys. Rev. Lett. 105, 015301 (2010).

  9. We thank the referee for pointing this out. We have changed the text in Sec 4.2.in the updated submission, it now reads "A key question is how much of the observed fluctuations and their correlations are fundamentally quantum, especially in systems with finite temperatures".

  10. We have changed “mean occupation number” to “mean Fourier spectrum” or “mean Fourier coefficients” in the relevant places throughout the manuscript.

  11. We have changed the notation for momentum modes of the relative phase to $q$ instead of $k$ in the updated manuscript. To incorporate this, we also changed our symbol for the sine-Gordon scaling factor $\lambda_T/l_j$ from $q$ to $\chi$ in all the relevant places throughout the manuscript.

  12. In the updated manuscript, we have changed the scaling factor in the definition to $1/\sqrt{L}$ and we have adjusted the scale in Fig. 8 and Fig. 21.

  13. We have provided more context for the similarities and differences between quantities in Sec. 4.1 and Sec. 4.4 in the updated submission.

  14. We thank the referee for pointing this out. We have added the following sentence in the text: “We note that this effect is due to dynamics in the high momentum modes which goes beyond the correction in Eq. (9). Indeed, for $t = 15\; \rm ms$ the expansion length scale is $\ell_t = \sqrt{\hbar t/(2m)} = 2.3 \mu m$ giving a momentum cutoff $q \sim \ell_t^{-1} \approx 0.43 \mu m$ which is smaller than the typical momenta where this oscillation is observed.”

  15. We understand the referee’s reasoning, however we would still like to maintain our use of “many-body problem” because the same principle of calculating higher-order correlation functions can be generalized to other contexts which are not necessarily described by classical sine-Gordon fields.

  16. In Fig. 11, considerable modification arises mostly for short length scales, e.g. in the text $z_3 = -z_4 = 5.5$ microns. We have explained in the text how the disconnected fourth order correlation can be considerably modified when probing short-distance. This is because within the second order correlator, the systematic phase shift introduces an “expanding cross region” at the center (see Fig. 9e-f). If we set our position (e.g. $z_3$ and $z_4$ to lie within this cross region, then the slice of fourth order correlation can be dominated by systematic effects. Meanwhile, if we choose far enough $z_3$ and $z_4$, then the deviation is found to be modest (Fig. 11d-f).

  17. We have included the value of $\chi = 3$ in the captions of Figs. 10 - 11.

  18. We thank the referee for pointing this out. We agree that “image processing” can be misleading given that we are considering physical effects arising from imaging. For this reason, we have changed the section title from “The effects of image processing” to “The effects of imaging”. In a few places where it is clear that we are referring to specific simulation implementations of such imaging effects, we keep the terminology “image processing”.

Lastly, we include a list of other important changes which are not in direct response to specific points from referees, but are made to improve presentation of the manuscript: - The first two paragraphs of the introduction have been slightly rewritten for clarity - Eq. (9) has been slightly modified, with some of the constant prefactors absorbed into the definition of $\ell_t$. More importantly, the sign of the second term is changed into negative. - (rederivation to correct for a sign error; other results unchanged) Theoretical derivation in Appendix C has been updated to include the fourth order Taylor expansion, which is important for getting the right constant prefactor in the second term of Eq. (9). - The title of Sec. 4.1 is changed from “Two-point phase correlation function” to “Vertex correlation functions” to distinguish it from the two-point correlation function discussed in Sec. 4.5. The relevant inline text and figure caption have been changed accordingly. - In Sec. 4.3 last paragraph, we added “To resolve such an effect, one must resolve fluctuation with a length scale comparable to the length scale of TOF dynamics $\ell_t = \sqrt{\hbar t/(2m)}$. Thus, this effect might not be captured by present experiments and by our perturbative treatment in Eq. (9). However, our numerical results point to the necessity of calibrating the results of dynamical propagation of velocity-velocity correlation such as in Ref. [19] to the measurement background in future experiments with enhanced resolution.”

---

## Round 1 · Author Response

Dear editor,

Thank you for your reply, and for facilitating the review process, especially given the unfortunate delay with the second referee. We appreciate your efforts and are grateful for the positive report we received. We have carefully considered and addressed all the suggestions provided by the current referee. We look forward to the next stage of the review, and hope that the second referee's feedback will be acquired promptly.

We very much appreciate the option to transfer the manuscript to your Core journal, but at this stage we would prefer to proceed with the review process for Scipost Physics. We would however be open to reconsidering the transfer option, if there are further significant delays in securing the second referee.

On behalf of the authors,
Nelly

---

## Round 1 · List of Changes

• On page 4, below Eq. (4), we changed ”interference peaks” to ”interference peaks amplitudes” as suggested by the referee.
• In the caption of Fig. 3, we fixed a typo ”amd” to ”and” as pointed out by the referee.
• We changed the histogram style of Fig. 6 and (Fig. 19 in the Appendix) for readability as suggested by the referee. We also fixed a typo 75 (nK) → 75 nK in the caption.
• We revised Sec. 4.4 to incorporate a more comprehensive discussion on the origin of oscillation in the extracted Fourier spectrum.
• In the second paragraph of Sec. 5, we fixed a repetition mistake "noise noise" into "noise".
• Still in the second paragraph of Sec. 5, we changed ”For example, as the atomic cloud scatters light, it receives a momentum transfer which can lead to diffusion of the atoms in the imaging plane and in absorption imaging, the incoming light is in the imaging direction,
which may push the image out of focus.” into ”For example, as the atomic cloud scatters light, it receives a momentum transfer which can lead to diffusion of the atoms in the imaging plane. Moreover, in absorption imaging, the incoming light is in the imaging direction,
which may push the image out of focus.”.
• On page 17, below Fig. 12, we changed ”In our study” to ”In our numerical study”.
• We changed Ref. [45] from ”In-preparation” to the reference to our recent arXiV preprint.
• We modified Eqs. (20)-(22) and the associated discussion in Appendix A to fix a minor algebraic mistake that does not change our main results.
• We added an extra factor of (1-0.5\partial_\eta^2 \phi_+)^{-1} into Eq. (40) in Appendix C.
• As suggested by the referee, we have changed the style of histograms in Fig. 6 and the related figure Fig. 19 in the Appendix. We remove the transparent bars and only draw the edges of the histogram with different colors. We hope that this change can make it easier for the readers to discern the different histograms in both figures.

---

## Round 2 · Author Response

Dear Editor,
We received all the referee reports with thanks, and note their recommendation for acceptance in Scipost Physics. We have since responded with detailed comments to all referee reports, and have revised our manuscript in this resubmission. We hope this revision gains favorable approval from you and our referees.
Best, Nelly

---

## Round 2 · List of Changes

• The first two paragraphs of the introduction have been slightly rewritten for clarity
  • Eq. (9) has been slightly modified, with some of the constant prefactors absorbed into the definition of $\ell_t$. More importantly, the sign of the second term is changed into negative. Hence, the theoretical derivation in Appendix C has been updated to include the fourth order Taylor expansion, which is important for getting the right constant prefactor in the second term of Eq. (9). Other results are unaffected by this change.
  • Changed title of Sec. 4.1 from “Two-point phase correlation function” to “Vertex correlation functions” to distinguish it from the two-point correlation function discussed in Sec. 4.5. The relevant inline text and figure caption have been changed accordingly.
  • Added in Sec. 4.3 last paragraph, “To resolve such an effect, one must resolve fluctuation with a length scale comparable to the length scale of TOF dynamics $\ell_t = \sqrt{\hbar t/(2m)}$. Thus, this effect might not be captured by present experiments and by our perturbative treatment in Eq. (9). However, our numerical results point to the necessity of calibrating the results of dynamical propagation of velocity-velocity correlation such as in Ref. [19] to the measurement background in future experiments with enhanced resolution.”
  • Added further elaboration in to discussions in the first paragraph of Sec. 6
  • Cited Ref [29] in response to referee’s comment
  • Changed the sentence on page 2, “If the trap is switched off rapidly, ... free evolution” into “If the trap is switched off rapidly, the initial tight confinement in transverse directions leads to rapidly expanding density, which allows one to neglect the effect of interactions during the expansion. Consequently, the dynamics are well approximated by a quench into free evolution.”
  • Removed the following sentence below Eq. (7) “The above implies that, at least up to the second order, longitudinal expansion does not change the functional relationship between $\rho_{\rm TOF}(x,z,t)$ and $\phi_-(z)$. This demonstrates the robustness of the transversal fit formula; nevertheless, longitudinal expansion still influences the extracted fit parameters. “ and replace it with “The form of Eq. (7) is expected from Eq. (3). Using our integrand expansion technique, we are able to express the fit parameters in terms of in-situ field variables.”
  • Removed sentence in Fig. 3 of the updated submission. “Numerical errors have been accounted for by subtracting the phase error using the transversal model in both encoding and decoding.”
  • Added “thermal state of the sine-Gordon Hamiltonian….” below the title of section 4
  • Text in Sec 4.2 updated.
  • Changed “mean occupation number” to “mean Fourier spectrum” or “mean Fourier coefficients” in the relevant places.
  • Changed the notation for momentum modes of the relative phase to $q$ instead of $k$, and also changed the symbol for the sine-Gordon scaling factor $\lambda_T/l_j$ from $q$ to $\chi$ in all the relevant places throughout the manuscript.
  • Changed the scaling factor in the definition to $1/\sqrt{L}$ and we have adjusted the scale in Fig. 8 and Fig. 21.
  • Provided more context for the similarities and differences between quantities in Sec. 4.1 and Sec. 4.4.
  • Added the following sentence in Sec. 4.4: “We note that this effect is due to dynamics in the high momentum modes which goes beyond the correction in Eq. (9). Indeed, for $t = 15\; \rm ms$ the expansion length scale is $\ell_t = \sqrt{\hbar t/(2m)} = 2.3 \mu m$ giving a momentum cutoff $q \sim \ell_t^{-1} \approx 0.43 \mu m$ which is smaller than the typical momenta where this oscillation is observed.”
  • Included the value of $\chi = 3$ in the captions of Figs. 10 - 11.
  • Changed the section 5 title from “The effects of image processing” to “The effects of imaging”.

---

## Editorial Decision

published